# MARPLE: A Benchmark for Long-Horizon Inference

**Emily Jin**[*]     **Zhuoyi Huang**[*]     **Jan-Philipp Fränken**     **Weiyu Liu**
**Hannah Cha**     **Erik Brockbank**     **Sarah Wu**     **Ruohan Zhang**
**Jiajun Wu**     **Tobias Gerstenberg**

Stanford University

## Abstract

Reconstructing past events requires reasoning across long time horizons. To figure out what happened, humans draw on prior knowledge about the world and human behavior and integrate insights from various sources of evidence including visual, language, and auditory cues. We introduce MARPLE, a benchmark for evaluating long-horizon inference capabilities using multi-modal evidence. Our benchmark features agents interacting with simulated households, supporting vision, language, and auditory stimuli, as well as procedurally generated environments and agent behaviors. Inspired by classic "whodunit" stories, we ask AI models and human participants to infer which agent caused a change in the environment based on a step-by-step replay of what actually happened. The goal is to correctly identify the culprit as early as possible. Our findings show that human participants outperform both traditional Monte Carlo simulation methods and an LLM baseline (`GPT-4`) on this task. Compared to humans, traditional inference models are less robust and performant, while `GPT-4` has difficulty comprehending environmental changes. We analyze factors influencing inference performance and ablate different modes of evidence, finding that all modes are valuable for performance. Overall, our experiments demonstrate that the long-horizon, multimodal inference tasks in our benchmark present a challenge to current models. Project website: `https://marple-benchmark.github.io/`.

## 1   Introduction

Long-horizon inferences are critical for solving "whodunit" problems in our every day lives. For example, we may wonder, "Who left the fridge open?", "Who spilled the food?", or "Who turned on the light?" To find out what happened and who did it, humans rely on an intuitive understanding of the physical world and how people interact with their environment to pursue their goals. Importantly, humans readily combine evidence across sensory modalities to figure out what happened [14, 40]. Long-horizon inferences are critical for solving "whodunit" problems in our every day lives. For example, we may wonder, "Who left the fridge open?", "Who spilled the food?", or "Who turned on the light?" To find out what happened and who did it, humans rely on an intuitive understanding of the physical world and how people interact with their environment to pursue their goals. Importantly, humans readily combine evidence across sensory modalities to figure out what happened [14, 40].

Developing AI models capable of performing such long-horizon reasoning and event reconstruction from multimodal information is critical for bridging the gap between human and machine intelligence. While the field of AI has developed increasingly powerful, general-purpose inference models [30, 38], the ability of these models to solve long-horizon inference problems, such as reasoning about "whodunit" scenarios, remains unclear. Existing benchmarks for evaluating inference capabilities focus on problems that require reasoning over short time horizons about physical events [3, 27] and over agent behaviors [28, 33]. In addition, they focus on visual stimuli, with only recent ones supporting language and audio [19, 21]. However, these benchmarks lack coverage of long-horizon,

---

[*]Equal contribution.

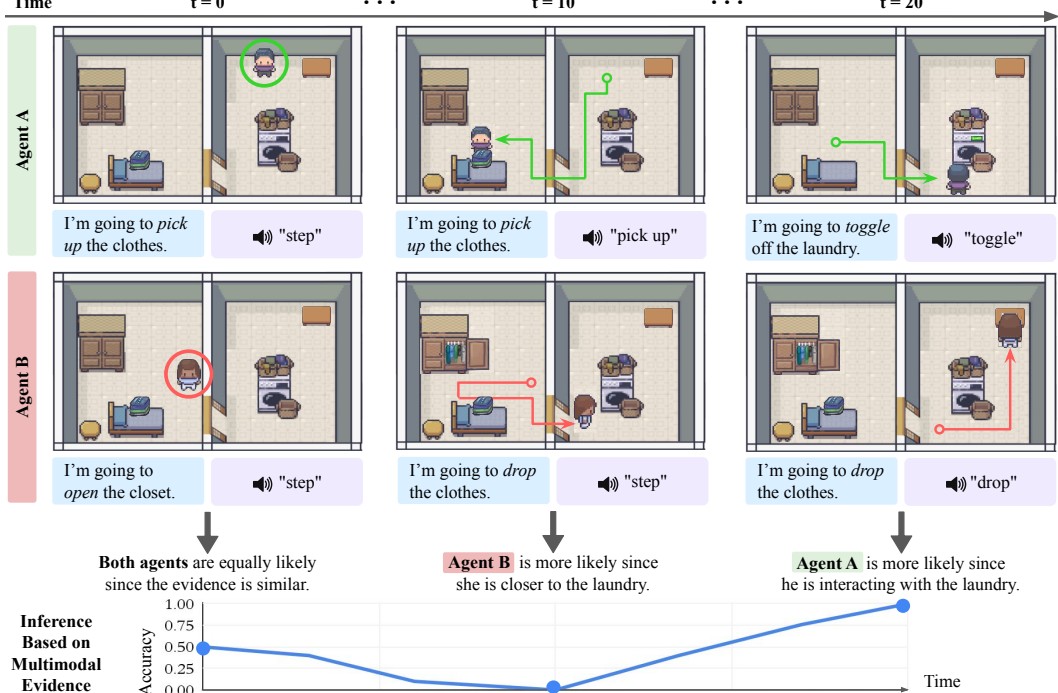

Figure 1: Illustrative example of an inference task in MARPLE: a "whodunit"-inspired benchmark for long-horizon inference. Given a query state change, the challenge is to decide which agent caused the change by leveraging visual, text, and/or audio evidence of both agents A and B up to some timestep $t$. The inference accuracy, probability of choosing the correct agent, is calculated at every timestep and used to evaluate performance.

multimodal inference in complex, everyday scenarios, a setting that is crucial for evaluating human-like reasoning abilities.

We propose MARPLE (in reference to Agatha Christie's Miss Marple) – a benchmark for long-horizon inference based on multimodal evidence. The main goal of MARPLE is to evaluate a model's ability to answer "whodunit"-style questions in daily household scenarios, such as "Who turned on the laundry?", by leveraging visual, text, and/or audio evidence. This inference problem challenges one to select the correct agent from two potential suspects, based on prior knowledge about agent behaviors and the state of the environment, as shown in Fig. 1.

In addition, we provide diverse training and inference data, along with well-defined evaluation metrics for our inference tasks. To systematically generate data, MARPLE builds upon the Mini-BEHAVIOR simulator [22], which simulates semantically rich daily activities in procedurally generated household Gridworld environments. We extend Mini-BEHAVIOR to support autonomous agents using hierarchical planners, enabling them to interact with the environment and generate multimodal evidence (vision, language, and audio). As a Gridworld, MARPLE facilitates the development of models focused on understanding high-level agent behavior, with the benefits of fast prototyping and training.

Using MARPLE, we benchmark two baselines against human performance. The **first baseline** uses traditional Monte Carlo tree search with learned agent models, while the **second baseline** uses a language model (GPT-4). To provide a comparison standard, we also conduct a behavioral study with **human participants**. Our findings reveal that that both baselines fall short in long-horizon, multimodal inference tasks compared to humans. The first baseline struggles to accurately predict future states and generalize to new environments, while the second one has difficulty reasoning about changes in the environment. Overall, we make the following key **contributions**:

1. We introduce a Gridworld simulator to procedurally generate household environments and diverse agent behaviors that yield multimodal evidence (visual, auditory, and language);

2. Using our simulator, we propose a set of long-horizon inference tasks for a) machine learning research on event reconstruction and multimodal reasoning and b) cognitive science

Table 1: Comparing MARPLE with other visual reasoning, causal reasoning, and cognition-inspired benchmarks. MARPLE is long-horizon, high-level, and with multimodal (vision, text, audio) support. *Time* refers to average stimuli length, *ecological* refers to object diversity, and *controlled generation* refers to annotated data generation.

| Benchmark | Time (seconds) | Video | Text | Audio | Ecological | High-Level Reasoning | Physical Realism | Controlled Generation | Cognition Inspired |
|---|---|---|---|---|---|---|---|---|---|
| CLEVR | - | - | ✓ | - | - | - | - | ✓ | - |
| MovieQA | 202.7 | ✓ | ✓ | - | ✓ | ✓ | - | - | - |
| TGIF-QA | 1.6 | ✓ | ✓ | - | ✓ | - | - | - | - |
| TVQA+ | 7.2 | ✓ | ✓ | - | ✓ | - | - | - | - |
| AGQA | 30 | ✓ | ✓ | - | ✓ | ✓ | - | - | - |
| MultiPLY | - | - | ✓ | ✓ | ✓ | - | - | ✓ | - |
| IntPhys | 7 | ✓ | - | - | - | - | ✓ | ✓ | - |
| Galileo | - | ✓ | - | - | - | - | ✓ | ✓ | - |
| CATER | 12.5 | ✓ | - | - | - | ✓ | ✓ | ✓ | - |
| CoPhy | 6 | ✓ | - | - | - | - | ✓ | ✓ | - |
| CRAFT | 10 | ✓ | - | - | - | - | ✓ | ✓ | - |
| CLEVRER | 5 | ✓ | - | - | - | - | ✓ | ✓ | - |
| ComPhy | 5 | ✓ | - | - | - | - | ✓ | ✓ | - |
| CLEVRER-Humans | 5 | ✓ | - | - | - | - | ✓ | ✓ | ✓ |
| AGENT | 15.4 | ✓ | - | - | - | - | - | ✓ | ✓ |
| BIB | 55 | ✓ | - | - | - | ✓ | - | ✓ | ✓ |
| PHASE | 17.5 | ✓ | - | - | ✓ | ✓ | ✓ | ✓ | ✓ |
| MMToM-QA | 63.4 | ✓ | ✓ | - | ✓ | ✓ | - | ✓ | ✓ |
| **MARPLE** | **52.5** | ✓ | ✓ | ✓ | ✓ | ✓ | - | ✓ | ✓ |

    research on the processes underlying human inference in complex scenarios. We also provide pre-collected datasets and evaluation metrics;

3. Lastly, we benchmark the performance of machine learning methods (Monte Carlo simulation and LLM) and human experts on the inference tasks.

## 2 Related Work: Cognition-Inspired AI Inference Benchmarks

Understanding how humans reason about causal relationships remains an active research area in cognitive science [31, 34]. To model this process, prior works have developed various frameworks, including the force dynamics model [37], mental models [16, 23], causal models [17, 35], and counterfactual simulation models [12, 13]. These frameworks provide insights into the cognitive mechanisms that underlie human inference abilities.

Inspired by human cognition, many machine learning benchmarks focus on problems that require the ability to reason about agents' interactions with their environment. These benchmarks emphasize different types of inference problems, such as reasoning about physical events [1, 3, 7, 15, 27, 45], agent behaviors [10, 11, 33], and multi-agent social behaviors [28]. While most of these benchmarks rely on visual stimuli, some recent ones support multimodal stimuli [19, 21], integrating both audio and vision [14]. Furthermore, several of these benchmarks provide human-annotated judgments and performance baselines [27, 28, 33], which are helpful for assessing the performance gap between humans and machines.

To address the inference problems in MARPLE, inference models must leverage knowledge about both the agents and the world. When these models are unknown, they can be learned from training data. The agent model allows the inference model to predict agent goals or actions, which can be learned through imitation learning [20, 32]. Meanwhile, the world model helps predict the consequences of taking an action from a given state. Recently, significant advancements in AI inference abilities have been made by machine learning-based models, such as large language models (LLMs) [36, 44], especially when combined with traditional search methods [43]. Our work presents and analyzes the performance of both a traditional search-based approach and an LLM-based method.

Despite progress in existing benchmarks, they primarily focus on short-term reasoning or single-modality stimuli, limiting their ability to evaluate models' performance in more complex, real-world scenarios. Our benchmark, MARPLE, addresses these shortcomings by providing a comprehensive framework for evaluating whether recent inference methods can solve long-horizon, multimodal inference tasks. In doing so, MARPLE aims to support the development of more robust and human-like AI reasoning capabilities.

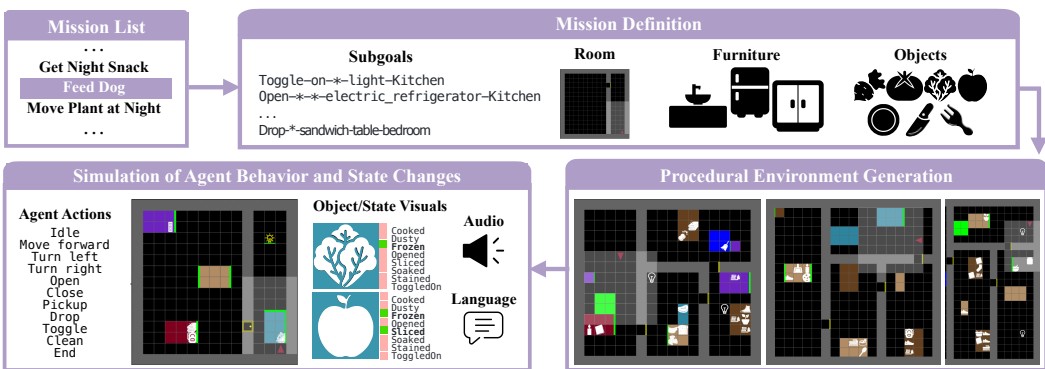

Figure 2: **MARPLE Household Simulator (backend)**. The simulator contains a list of pre-defined `Missions`, each mission consists of a list of `Subgoals`, and each subgoal is a representation of a `Action-State_change-Object-Furniture-Room` combination. Given the mission definition and corresponding environment configuration file, we can procedurally generate the environment.

## 3 MARPLE Benchmark

**Overview.** As shown in Table 1, MARPLE focuses on inference problems in long-horizon settings with multimodal support. It is highly configurable, with support for procedural generation of rich agent behaviors and diverse environment states at an abstract, semantic level. Specifically, MARPLE provides a variety of *inference scenarios* for "whodunit"-type questions, in which two agents, A and B, each perform a *mission*: a common household activity that humans perform in real life. To carry out a mission, an agent interacts with the environment, causing changes in the world and leaving evidence of its activity. A "whodunit" question is constructed by selecting a unique state that only appears in one agent's trajectory. For example, consider an inference scenario where agents A and B have completed the missions `do laundry` and `get snack`, respectively. A state that is unique to agent A is "laundry machine is on," so we pose the following question: "Who turned on the laundry?" To answer "whodunit" questions, models must leverage evidence in the form of multimodal observations from each agent's activity history. An example of the inference process is shown in Figure 1.

**Problem Formulation.** We formalize the inference problem using a Partially Observable Markov Decision Process (POMDP), denoted by the tuple $\langle \mathcal{S}, \mathcal{A}, \mathcal{R}, \mathcal{T}, \Omega, \mathcal{O}, \gamma \rangle$, where $\mathcal{S}$ is the state space, $\mathcal{A}$ is the action space, $\mathcal{R}$ is the reward function, $\mathcal{T}$ is the transition function, $\Omega$ is a set of observations, $\mathcal{O}$ is the observation function, and $\gamma$ is the discount factor. The state at time step $t$ is $s_t$, and visual, auditory, and language observations are denoted by $o_t = \{o_t^V, o_t^A, o_t^L\}$. The action space $\mathcal{A}$ consists of low-level agent actions, and an agent's action trajectory is determined by its mission. A mission is decomposed into a sequence of mid-level subgoals $g \in \mathcal{G}$, which are further decomposed into low-level actions. Each subgoal relies on the completion of past ones and is necessary for completing future ones, creating strong multi-step dependencies between the actions. We represent agent $i$'s behavior using a policy $\pi^i : \Omega \to \mathcal{A}$ that maps observations $\Omega$ to a probability distribution over actions in $\mathcal{A}$, while the transition function $\mathcal{T} : \mathcal{S} \times \mathcal{A} \to \mathcal{S}$ determines the effects of agent actions.

In each scenario, the objective is to infer whether agent A or B is more likely to have caused a particular query state (e.g., "laundry is on"). We formulate this as predicting the probability $P(s_T | \pi^i, o_{0:\tau})$ for agent $i$ at any intermediate time step $\tau$, where $s_T$ is the state in query, and $o_{0:\tau}$ are observations until time step $\tau$. Different instantiations of $o_{0:\tau}$ affect the horizon, and hence inference difficulty. For example, when $\tau = T$, inference is trivial.

Solving an inference scenario requires knowledge about the world model $\mathcal{T}(s'|s, a)$, observation model $\mathcal{O}(o|s)$, and policy $\pi^i(a|o)$ for both agents. The true agent policies are unknown to the inference model and need to be learned in a training stage. A training dataset of previous agent behaviors $\mathcal{D}_i = \{\zeta_1, \zeta_2, ..., \zeta_n\}$ is collected, where each trajectory $\zeta$ is a sequence of agent actions $\{a_0, a_1, .., a_T\}$ paired with observations $\{o_0, o_1, .., o_T\}$. We assume that agents can perform multiple missions, with their preferences for the missions represented as a prior distribution over all possible ones. For example, an agent might prefer to `get snack` with probability $0.8$, `pickup the plant` with probability $0.2$, and all other missions with probability $0$. When simulating the agent's trajectories, the missions are sampled according to their mission preferences.

**Evaluation.** In our problem setting, inference ability is measured by the probability of correctly choosing the agent responsible for the query state. We are interested in how much evidence is needed to make the correct inference: stronger models require less evidence and achieve high inference accuracy at earlier time points. Other factors that affect performance include inference scenario difficulty, environment complexity, agent behavior similarities, and inference horizon.

## 4   MARPLE Household Simulator

To generate our benchmark, we introduce the MARPLE Household Simulator, shown in Figure 2. The simulator supports a wide variety of complex scenarios and generates diverse data. It consists of two components: a multimodal environment simulator and a hierarchical agent planner. Our simulation environment is built on top of Mini-BEHAVIOR [22], which supports 20 household activities, fast simulation, and procedural generation of room layouts. By abstracting away low-level physical details, MARPLE enables researchers to efficiently prototype and evaluate their high-level, long-horizon inference models. Additional details about the simulator and computational resources are in Appendix D. Our simulator extends Mini-BEHAVIOR to support multimodal stimuli, procedural generation of diverse agent behaviors, and a human experiment user interface (UI).

**Multimodal Environment Simulator.** Our simulator additionally supports language and auditory stimuli. The *language* modality is a natural language description of the subgoal that the agent intends to perform next. For example, the subgoal `ToggleOn(light)` is described as *"I am going to toggle on the light."*. This modality offers insight into the agent's future intentions without revealing its mission until the ultimate subgoal. The challenge lies in effectively leveraging this information to understand how these intentions relate to the final state. We carefully constructed scenarios where the language modality helps to varying degrees. For example, in the scenario "Who picked up the snack?", language evidence reveals early on that agent A intends to "open refrigerator" while agent B intends to "pickup towel from closet." From this, a strong inference model should be able to reason that agent A is more likely to pick up the snack. On the other hand, consider the scenario "Who toggled on the laundry", where both agents share many subgoals. Agent A performs: "pickup clothes from bed", "open laundry", "drop clothes", "close laundry", "toggle-on laundry", while Agent B performs: "open closet", "pickup clothes from closet", "close closet", "open laundry", "drop clothes", "close laundry". In this case, language evidence only helps distinguish between agents at the end.

The audio modality is generated by mapping each possible agent action to a corresponding sound. This mapping is not one-to-one; for example, all navigation actions (left, right, forward) share the same *step* sound. Such audio evidence reveals partial information about the agent's low-level actions, which can be useful for resolving state uncertainty in inferential settings [14, 24, 40]. For example, consider the scenario "Who turned on the laundry?", where visual evidence reveals that Agent A is in the same room as the laundry, just 5 steps away, while Agent B is in a bedroom 20 steps away with the door closed. Based solely on this, one might infer that Agent A was the likely culprit due to proximity. However, if audio evidence reveals a long sequence of steps or a door closing, it might suggest that Agent B was responsible. Leveraging audio to infer what happened presents a challenging research direction. For details on language and auditory simulation generation, see Appendix B.

**Procedural Generation of Agent Behaviors.** To generate agent behaviors, we use a hierarchical planner with high-, mid-, and low-level components, as illustrated in Figure 3. The high-level planner first selects a *mission* based on the agent's mission preferences, and the mid-level planner breaks the mission down into a sequence of subgoals. Each subgoal is defined by an *action*, *object*, and *state*. The low-level planner further decomposes each subgoal into a sequence of atomic actions to perform, which includes actions for navigation (turn left, turn right, and move forward) and the action specified by the subgoal itself. In particular, the low-level planner uses the A-star algorithm [18] to plan the shortest path to navigate to the subgoal position, perform the subgoal *action* on the *object*, and ultimately produce the desired *state*. When multiple

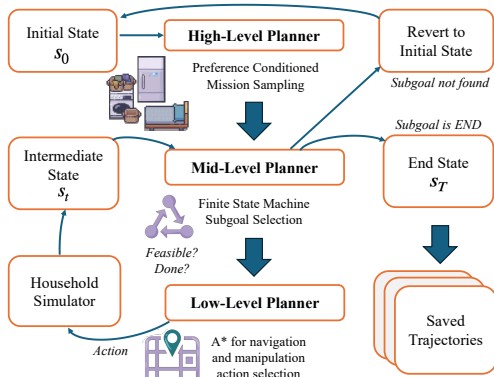

Figure 3: A hierarchical planner for procedural generation of agent behaviors. A high-level planner samples a mission, a finite state machine breaks it into subgoals, and a low-level planner determines actions.

optimal paths exist, the planner randomly selects one to introduce variability. This approach avoids unnecessary actions or random walks, ensuring that every action in the trajectory directly contributes to completing the mission. Our planner is able to generate large amounts of diverse, long-horizon agent trajectories based on the specified mission, subgoals, room layouts, and initial positions.

**Human Experiment User Interface.** Mini-BEHAVIOR's visualization is suitable for machine learning research but not human studies. Hence, we develop a more intuitive, aesthetically pleasing interface, as shown in Appendix K. This extension allows us to collect human data to establish performance baselines, as well as support future cognitive science experiments using MARPLE.

**Inference Scenarios and Dataset.** With these new features, we define the MARPLE Benchmark with ten diverse, long-horizon missions and provide both training and testing data. We construct various inference scenarios by combining missions and assigning mission preferences to each agent. In experiments, we demonstrate the simplest case by only having A perform one mission and B another. We pair up all 10 missions to define 5 distinct *inference scenarios* with a query state selected to be a meaningful subgoal unique to one agent, as shown in Table A.1. These 5 scenarios offer a manageable representation of the diversity and complexity offered by pairing missions. Details on the inference scenarios and selection process are provided in Appendix A.

For each mission, we provide a test dataset with 500 diverse agent trajectories, generated in 10 environments featuring different room layouts and object placements. We also provide two training datasets with 5000 trajectories each: one with 500 agent trajectories in each of the 10 test environments, and the other with one trajectory per 5000 procedurally generated environments. The test environments are unseen by models trained on the second dataset, enabling evaluation of generalization capabilities. These datasets offer diverse scenarios for training and evaluating inference models.

# 5 Inference Methods and Baselines

## 5.1 Simulation with Learned Agent Models

Our first inference method (Appendix E) uses Monte Carlo Tree Search (MCTS) [6] with learned agent policy models. At inference time, this method performs Monte Carlo rollouts starting from time $\tau$, assuming that it has access to the ground truth world model (provided by the simulator). An agent-specific policy for agent $i$, $\pi^i : \Omega \to \mathcal{A}$, is first learned through imitation learning from a dataset of past agent behaviors. We perform $m$ Monte Carlo rollouts for each agent $i$ starting from the current state $s_\tau$ and observation $o_\tau$, and the model predicts agent action $a_\tau$ using the learned policy model. Then, the predicted action is passed to the simulator to query for $s_{\tau+1}$ and $o_{\tau+1}$, and the model predicts the next action. The probability of reaching the query state $s_T$, given by $P(s_T|\pi^i, o^i_{0:\tau})$, corresponds to the fraction of the $m$ sampled rollouts that reach $s_T$. Assuming Boltzmann rationality [2, 29], normalized predictions are obtained by applying a softmax function to the probability for each agent. For example, for agent A, the prediction is ($\eta = 5$ is the temperature parameter): $P(A) = \exp\left(\eta P(s_T|\pi^A, o^A_{0:\tau})\right) / \exp\left(\eta P(s_T|\pi^A, o^A_{0:\tau})\right) + \exp\left(\eta P(s_T|\pi^B, o^B_{0:\tau})\right)$. Now, we discuss four variants of this baseline, each of which uses different types of observations.

**Vision-Only Model.** The first variant learns to predict the next low-level action $a_{t+1}$ given the current visual observation $o^V_t$. It uses a vision transformer [8] as an encoder and a policy head that outputs a probability distribution over all possible actions $P(a|o^V_t)$. The network is trained using supervised learning, i.e., through behavioral cloning [32].

**Audio-Augmented Model.** Our second implementation leverages both visual $o^V_t$ and audio $o^A_t$ observations. Audio information is used here in a limited setting to improve the prediction accuracy of the first action in the rollout, as it reveals partial information about the agent's next low-level action. We first obtain a predicted action distribution from the vision-only model, and then leverage audio evidence to refine the distribution. We then obtain the probability of the next action being an action $a$, conditioned on the visual and audio observations, by using Bayes' rule: $P(a|o^V_t, o^A_t) \propto P(o^A_t|a)P(a|o^V_t)$, where the probability $P(a|o^V_t)$ is predicted by the vision-only model, and $P(o^A_t|a)$ is computed using a mapping from the action to the audio observation that is given.

**Language-Conditioned Model.** The third variant uses language observations $o^L_t$, which reveal information about the subgoal that the agent is aiming to achieve at time $t$. Intuitively, the intended subgoal reveals future information that will improve low-level action prediction accuracy. At time $t$, the language-conditioned model predicts the next low-level action $a_t$ by conditioning on both the visual observation $o^V_t$ and the subgoal revealed by the language observation $o^L_t$.

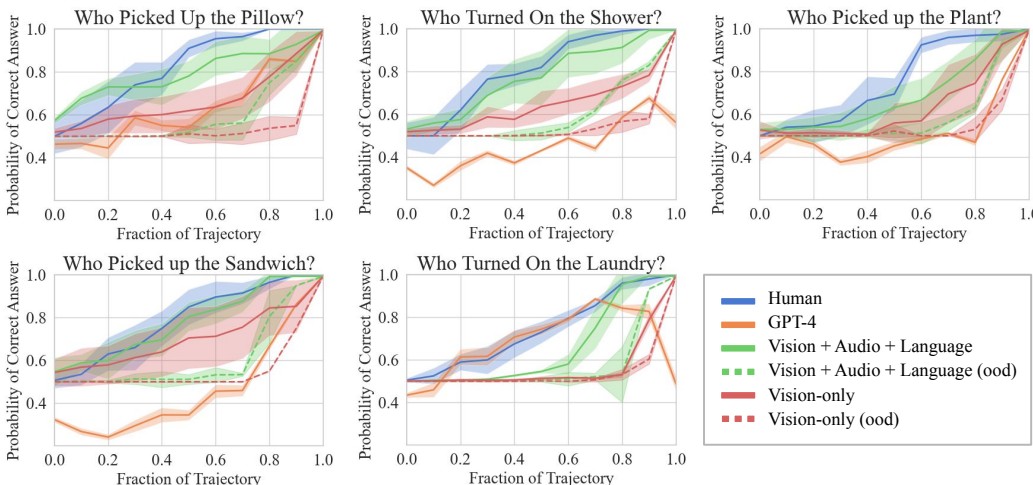

Figure 4: Performance for each baseline across scenarios. Results are included for the simulation baseline trained both in-distribution and out-of-distribution (ood). Inference scenarios are presented in order of increasing difficulty from left to right, top to bottom. Error bands correspond to 95% CI intervals across tested trajectories.

**Audio-Augmented Language-Conditioned Model.** The final variant uses observations from all three modalities – vision, language, and audio. At test time $t$, this variant uses the language-conditioned model to predict the next action $a_t$, conditioned on both the visual $o_t^V$ and language observation $o_t^L$. Audio evidence $o_t^A$ is then leveraged to refine the distribution over possible actions.

## 5.2 Additional Baselines

**LLM.** For our second class of baselines, we ask state-of-the-art large language models to predict which agent is more likely to have caused the query state, given visual observations of both agents at two consecutive timesteps, $o_{\tau-1}^V$ and $o_\tau^V$. We benchmark `GPT-4-0613` with a standard zero-shot "let's think step-by-step" prompt [39, 42] as our primary LLM baseline. In addition, we evaluate the performance of top open-source models, `Llama-3.1-8B-Instruct` [9] and `Qwen2-7B-Instruct` [41], chosen due to their large context length.

To perform the inference task, these LLMs must reason about changes in consecutive states and consider how the agent may reach the query state $s_T$. Both the evidence and query states are provided to the model using a standard scene graph representation [25], containing a set of nodes and directed edges. Each node represents an agent or object, along with the states of that entity (e.g., a drawer is open). The directed edges represent physical relations between entities, e.g., "onTop" (object-object relation) and "inRoom" (object-room relation). See Appendix G for a simplified prompt. For a more comprehensive analysis, we also benchmark `GPT-4` with in-context learning on select inference scenarios. We modify our zero-shot prompt and include examples from two other trajectories. Each example contains the inference answer and scene graphs of the current, previous, and query states of both agents at the same time step.

**Human Baseline.** As a third baseline, we run an experiment with two human experts. Each participant is provided with a habituation phase, in which they are familiarized with MARPLE domain knowledge, the inference setup, and a few examples of agent trajectories. During experiments, participants answer the inference question, given side-by-side visual observations of agent trajectories, presented one step at a time from $t = 0$ to $\tau$ (as in Figure K.1). This allows participants to build an incremental understanding of agent trajectories and compare agent behaviors within the scenario.

## 6 Experiments and Results

### 6.1 Benchmarking Model Performance in Long-Horizon Inference Scenarios

For each inference method and baseline, we run experiments on all five inference scenarios shown in Table A.1. We test on 10 randomly generated environments of each inference scenario, resulting in 50 total trials (see Appendix D for more details). For each trial, we ask the model to answer the inference question and obtain its inference accuracy given evidence at various time steps, namely

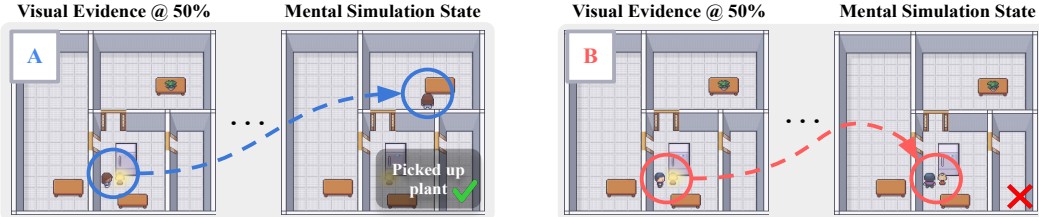

Figure 5: Example rollouts performed by our simulation model, starting from the initial state to possible future states. For agent A, this rollout reaches the inference state: `Pickup(plant)`.

$\tau = 0, \tau = T/10, ..., \tau = T$. The inference problem becomes easier at later time steps, as more evidence is revealed, and the inference horizon decreases. Thus, we expect accuracy to increase as $\tau$ increases. We are especially interested in how much evidence is required to choose the correct agent.

For our MCTS baseline, we focus on two variants: vision-only and audio-augmented language-conditioned. In this setup, each agent always performs one mission, and the agent models are trained on a dataset of agent trajectories for that mission. The dataset contains 500 trajectories in each of the 10 environments seen at test time. The number of rollouts is set to be $m = 100$. For our second baseline, we use `GPT-4-0613` at temperature $T = 0.5$ using $n = 10$ completions for each API call.

**Main Results.** Our key results are summarized in Figure 4. Across all five inference scenarios, the accuracies of all baselines increase over time and eventually converge at the end of the trajectory (except `GPT-4`, as discussed below). Our evaluation, however, is centered on how early the methods are able to make the correct inference, rather than on convergence itself. With this in mind, we see that MARPLE presents a challenging benchmark for all baselines. Overall, human participants provide a strong upper bound on performance, even without extensive prior knowledge about the agents' preferences and past behaviors. Humans consistently outperform all models, achieving higher accuracies with less evidence and demonstrating stronger robustness.

**Analysis of Simulation Methods.** When contrasting simulation methods (vision-only and vision+audio+language) with `GPT-4`, we observe that simulation-based models generally achieve higher accuracy and always converge to $1.0$ by the end of the trajectory. This highlights the benefit of explicitly modeling agent behaviors and performing step-by-step simulations. As a concrete example, we examine an instance of the scenario: "Who picked up the plant?" Evidence shown $50\%$ into the trajectory reveals that the two agents are in the same state – next to the turned-on light – as shown in Figure 5. In this case, `GPT-4` doesn't make the correct inference, as it only considers the evidence at the current and last time steps. Meanwhile, the simulation baseline achieves a 0.9 accuracy. The state `ToggledOn(light)` is a meaningful one that always occurs before `Pickup(plant)`, and the simulation baseline leverages its knowledge of agent behaviors to successfully estimate future states.

**Analysis of LLM Performance.** While `GPT-4` performs competitively on some inference scenarios, `GPT-4` fails to converge on two in particular: "Who turned on the shower?" and "Who turned on the laundry?". In Appendix I, we provide additional results of `GPT-4` with in-context learning (ICL) applied to these two scenarios where it does not converge. We find that although ICL improves `GPT-4`'s performance, it still struggles to converge. An analysis of `GPT-4`'s chain-of-thought reasoning reveals that the model was biased toward changes in agent states, such as position, direction, or whether the agent was carrying an object. We speculate that this prevented `GPT-4` from converging for these two tasks because their query states were only reflected as a change in the environment state and *not* the agent state. By contrast, in the other three scenarios, the agent was holding an object in the query state, making it easier for `GPT-4` to infer the correct answer. Examples of zero-shot and ICL reasoning mistakes are provided in Appendix J.

Additionally, we provide the results of the open-source LLMs `Llama-3.1-8B-Instruct` and `Qwen2-7B-Instruct` in Appendix H. Both demonstrate lower perform than `GPT-4` and exhibit similar inconsistencies in performance, including difficulties in comprehending changes in the environment and ultimately failing to converge.

Despite the abilities of these LLMs to perform strong general reasoning, their failure modes reveal important opportunities for future work that better leverage in-context examples [5] or additional scaffolds [4] to study language models on our benchmark.

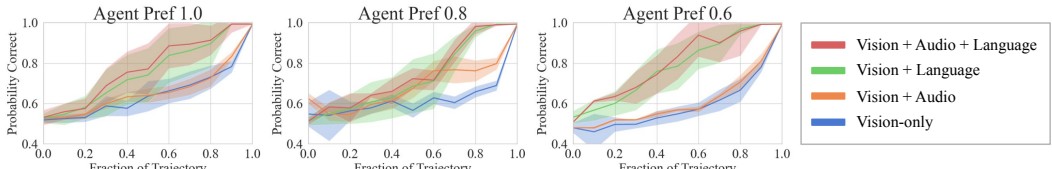

Figure 6: Performance for all variants of the simulation baseline, for one inference scenario: "Who turned on the shower?". The error bands correspond to 95% CI intervals across test trajectories.

**Analysis of Human Performance.** Humans consistently outperform the other baselines, on average reaching 0.8 accuracy given only 48% of the evidence. Even without significant training, humans require 10% and 47% less evidence than the best MCTS variant in-distribution and `GPT-4` (Table 2).

## 6.2 Benchmarking Generalization Capabilities of Simulation Models

We run additional experiments on all five inference scenarios to evaluate the generalization capabilities of the simulation approach. We train models under two settings: one with trajectories in the same 10 environments as the test set, and the other using procedurally generated environments and tested in 10 unseen environments. While the models perform well in distribution, they struggle to generalize to novel environments (Table 2). Even the vision + audio + language variant, the strongest MCTS method, suffers a significant performance drop in unseen environments (Figure 4). This is primarily because the learned agent model does not generalize well to novel environments, leading to decreased accuracy in action prediction and rollouts. In sharp contrast, humans achieve strong performance even without prior training. As shown in Table 2, the performance gap between humans and the best simulation method increases from 10% to 33% less evidence out-of-distribution, highlighting significant room for improvement in building robust and generalizable inference models.

## 6.3 Benchmarking in Multimodal Settings

We now study how incorporating multimodal observations can improve the simulation model's performance. We conduct experiments on the four variants of the simulation baseline: vision-only, audio-augmented, language-conditioned, and audio-augmented and language-conditioned. The results for "Who turned on the shower?" are shown in Figure 6. While language seems more valuable than audio in our setting, the baseline using all three modalities consistently outperforms the others. This suggests that audio and language provide different signals and are both beneficial.

**Effect of Audio Evidence.** In all settings, audio evidence slightly improves performance over the vision-only model, as correctly predicting the current action results in a more accurate distribution of the rollouts. This demonstrates the benefit of including audio evidence, but note that the benefits are limited under this setup as we only leverage one timestep of audio evidence for one action prediction.

**Effect of Language Evidence.** We find that the language-conditioned model significantly outperforms other baselines and stays consistent even when others' performances decrease. As expected, knowing the subgoal leads to more accurate action prediction. When evaluated on the inference trajectories, the language-conditioned policy achieves 0.92 accuracy, as compared to 0.86 for the vision-only policy. This advantage is critical for boosting performance in long-horizon rollouts due to compounding errors and is even more salient under challenging inference settings, as discussed next.

## 6.4 Additional Benchmarking Experiments

In contrast to our primary experiments, where we assume that each agent is dedicated to a single mission, this time, we allow agents to undertake both their own mission and the other's. We vary agent preferences to be 1.0, 0.8, and 0.6 for their own mission and 0.0, 0.2, and 0.4 for the other, respectively. We use the inference scenario where the agents perform `feed dog` and `do laundry` due to the substantial differences between the two missions. The distinct subgoals of the two agents

Table 2: Evidence needed for the baselines to achieve a 0.8 inference accuracy, quantified by the fraction of trajectories shown. Humans consistently make more accurate predictions earlier, particularly out-of-distribution.

|  | Human | Vision + Audio + Language | Vision + Language | Vision + Audio | Vision-Only | LLM |
|---|---|---|---|---|---|---|
| In-Distribution ↓ | 0.48 | 0.58 | 0.64 | 0.80 | 0.85 | 0.95 |
| Out-of-Distribution ↓ | 0.48 | 0.81 | 0.85 | 0.91 | 0.92 | 0.95 |

result in divergent agent behaviors when each has a 1.0 preference for their primary mission. As agent preferences converge – such as 0.6 for their own mission and 0.4 for the other – agent behaviors become increasingly similar, thereby increasing inference difficulty.

**Effect of Agent Preferences.** As agent preferences converge and agent behaviors become more similar, we see that performance worsens for the vision-only and audio-augmented models. When agents have a preference of 1.0 for their primary missions, both models reach 0.6 inference accuracy when observing around $40\%$ of the trajectory. When the primary mission preferences are 0.6 though, model performance decreases. The audio-augmented and vision-only models require evidence up to $70\%$ and $85\%$ of the whole trajectory, respectively, to reach the same accuracy of 0.6.

## 7 Limitations and Conclusion

**Limitations.** While our benchmark is well-suited for studying high-level, commonsense reasoning and inference, it has several limitations. First, our simulation environment is a GridWorld, which lacks physical realism and is therefor not suitable for low-level physical reasoning tasks. Second, our language and audio stimuli are limited as they are generated from a defined library and mapping. In the future, we plan to enhance our simulator with free-form natural language descriptions and realistic audio renderings to create a more comprehensive and realistic testbed. Lastly, there are some limitations of our problem setup. For example, we focus on "whodunit" scenarios where the selected state is unique to one agent's trajectory, simplifying the inference problem. Future work could explore scenarios where the query state could potentially be caused by either agent. Furthermore, although our current setup involves only two agents, our simulator is capable of supporting multiple agents acting in parallel. However, it does not yet support agent interactions, which would introduce additional complexity to the inference process. Despite these limitations, our current setup remains challenging, as it requires models to understand diverse agent behaviors and generalize across various environments, making it a strong foundation for benchmarking inference methods.

**Conclusion.** We introduced MARPLE, a novel benchmark for evaluating long-horizon, multimodal inference capabilities. We find that current AI models, including Monte Carlo tree search and LLM methods fall short of humans in leveraging multimodal stimuli and performing long-horizon inference. We hope that MARPLE facilitates further AI and cognitive science research to bridge the gap between artificial and human cognitive abilities in complex, real-world inference scenarios.

## Acknowledgments and Disclosure of Funding

This work was in part supported by the Stanford Institute for Human-Centered Artificial Intelligence (HAI), NSF CCRI #2120095, RI #2338203, ONR YIP N00014-24-1-2117, and ONR MURI N00014-22-1-2740.

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

# Supplementary for:
# MARPLE: A Benchmark for Long-Horizon Inference

The MARPLE website is at: `https://marple-benchmark.github.io/`.

The appendix is organized as the following. In Appendix A, we present details about the benchmark and inference scnearios. In Appendix B, we present details about the hierarchical simulator used to generate multimodal evidence and trajectories. In Appendix C, we provide details about our dataset and access. In Appendix D, we provide details on the computational resources and experiment details. In Appendix E and Appendix F, we present implementation details and ablations for the simulation method. In Appendix G, we present the prompts used for `GPT-4`. In Appendix H and Appendix I, we provide additional results benchmarking open-source LLMs and `GPT-4` with in-context learning. We include analysis of `GPT-4` reasoning in Appendix J. Lastly, in Appendix K, we present details on the human experiments.

## A    MARPLE Benchmark: Inference Scenarios

### A.1    Overview

The MARPLE codebase can be found at `https://github.com/marple-benchmark/marple`. Our benchmark consists of 10 household missions paired to create a set of 5 inference scenarios, as shown in Table A.1. This provides a representative sample of the diversity and complexity possible by pairing missions.

Table A.1:  Five inference scenarios in our benchmark, defined in terms of the inference question, agent `A`'s mission, and agent `B`'s mission. For these tasks, agent `A` is always the answer to the inference question. The tasks are in order of increasing difficulty, which is determined based on the average inference horizon and similarity between the two missions.

| Inference Question | Agent A Mission | Agent B Mission | Avg. Horizon | Similarity |
|---|---|---|---|---|
| Who picked up the pillow? | Watch movie cozily | Watch news on TV | 15 | 0.19 |
| Who turned on the shower? | Take shower | Feed dog | 26.4 | 0.30 |
| Who picked up the snack? | Get snack | Clean living room table | 36.8 | 0.46 |
| Who picked up the plant? | Move plant at night | Get night snack | 43.9 | 0.61 |
| Who turned on the laundry? | Do laundry | Change outfit | 51.3 | 0.87 |

### A.2    Inference Scenario Setup

An inference scenario is defined by the missions performed by agents `A` and `B` and a query state. We provide details on the necessary components below:

**Missions.** We define 10 household missions: `Change Outfit`, `Clean Living Room Table`, `Do Laundry`, `Feed Dog`, `Get Night Snack`, `Get Snack`, `Move Plant at Night`, `Take Shower`, `Watch Movie Cozily`, `Watch News on TV`. These missions vary in the number of timesteps and types of actions. Each mission is defined by a list of subgoals, which we define next.

**Subgoals.** A mission's subgoal is a symbolic state that must be satisfied to complete the mission. It is represented as a dictionary with the keys "obj", "fur", "room", "pos", "action", "state", and "end_state." The "obj" and "fur" determine the target object type, "room" and "pos" describe the target location, and "action" is the action that the agent must perform on the target to result in the desired "state." The "state" is a tuple with the state name and boolean value, and "end_state" is `True` if the subgoal is the last one in the mission and `False` otherwise.

We provide an example of a mission and subgoal representation in Figure A.1.

**Inference Scenario.** To construct an inference scenario, we pair two missions (e.g., `do laundry` and `change outfit`) and select a query state unique to one agent (e.g., `Pickup(sandwich) = True`). The corresponding inference question is: "Which agent is more likely to have [state action] the [state object]?" For instance, if the query state is `Pickup(sandwich)`, the question would be: "Which agent is more likely to have `picked-up` the `sandwich`?"

### A.3    Inference Scenario Difficulty

We identify two key factors that affect the difficulty of an inference scenario: the average inference horizon and the similarity between the two missions.

```
Mission and Subgoal Representation

Example of a Mission:  list of subgoals
get_night_snack = [
    toggle-on-*-light-Kitchen,
    open-*-*-electric_refrigerator-Kitchen,
    pickup-*-sandwich-electric_refrigerator-Kitchen,
    close-*-*-electric_refrigerator-Kitchen,
    toggle-off-*-light-Kitchen,
    drop-*-sandwich-table-Bedroom
]

Example of a Subgoal:  tuple with subgoal name, subgoal dictionary
(
  ''toggle-on-*-light-Kitchen'',
  {
      ''obj'':  None,
      ''fur'':  ''light'',
      ''room'':  ''Kitchen'',
      ''pos'':  None,
      ''action'':  ''toggle'',
      ''state'':  [''toggleable'', 1],
      ''can_skip'':  False,
      ''end_state'':  False
  }
)
```

Figure A.1: Example of a mission and subgoal representation, for the mission: `Get Night Snack`.

**Inference Horizon**. The inference horizon is the number of steps that it takes for agent A to reach its inference state. As the inference horizon increases, difficulty increases because models must understand and predict more future steps. The uncertainty in predictions also compounds over time, leading to greater prediction errors and variation in possible outcomes.

**Mission Similarity**. An inference scenario becomes more challenging when the two agents have similar trajectories, which are largely determined by their missions' subgoals. Thus, we define the similarity between a pair of missions, $M_1$ and $M_2$, as follows:

$$\text{similarity}(M_1, M_2) = \frac{1}{1.5} \left( \frac{|M_1 \text{ subgoal actions} \cap M_2 \text{ subgoal actions}|}{|M_1| \text{ subgoal actions} \cup M_2 \text{ subgoal actions}|} + 0.5 \frac{|M_1 \text{ subgoal rooms} \cap M_2 \text{ subgoal rooms}|}{|M_1| \text{ subgoal rooms} \cup M_2 \text{ subgoal rooms}|} \right)$$

Our chosen set of inference scenarios represents a range of similarities, as shown in Table A.2.

Table A.2: Similarity of all possible pairs by combining the 10 missions. Of these pairs, the similarity ranges from 0.19 to 0.87. Our chosen set of inference scenarios is highlighted in blue, and they span a wide range of the similarity values to represent a range of difficulties.

| | change outfit | clean living room table | do laundry | feed dog | get night snack | get snack | move plant at night | take shower | watch movie cozily | watch news on tv |
|---|---|---|---|---|---|---|---|---|---|---|
| change outfit | 1.00 | 0.53 | 0.87 | 0.78 | 0.6 | 0.53 | 0.29 | 0.44 | 0.28 | 0.19 |
| clean living room table | 0.53 | 1.00 | 0.33 | 0.64 | 0.56 | 0.46 | 0.48 | 0.19 | 0.25 | 0.28 |
| do laundry | **0.87** | 0.33 | 1.00 | 0.46 | 0.56 | 0.74 | 0.64 | 0.71 | 0.64 | 0.56 |
| feed dog | 0.60 | 0.64 | 0.46 | 1.00 | 0.87 | 0.67 | 0.37 | 0.30 | 0.28 | 0.19 |
| get night snack | 0.64 | 0.56 | 0.56 | 0.87 | 1.00 | 0.78 | 0.61 | 0.61 | 0.37 | 0.29 |
| get snack | 0.53 | **0.46** | 0.74 | 0.67 | 0.78 | 1.00 | 0.64 | 0.61 | 0.55 | 0.46 |
| move plant at night | 0.29 | 0.48 | 0.64 | 0.37 | **0.61** | 0.64 | 1.00 | 0.35 | 0.55 | 0.60 |
| take shower | 0.44 | 0.19 | 0.71 | **0.30** | 0.42 | 0.61 | 0.35 | 1.00 | 0.70 | 0.62 |
| watch movie cozily | 0.28 | 0.25 | 0.64 | 0.28 | 0.37 | 0.55 | 0.55 | 0.70 | 1.00 | 0.19 |
| watch news on tv | 0.19 | 0.28 | 0.56 | 0.19 | 0.29 | 0.46 | 0.60 | 0.62 | **0.19** | 1.00 |

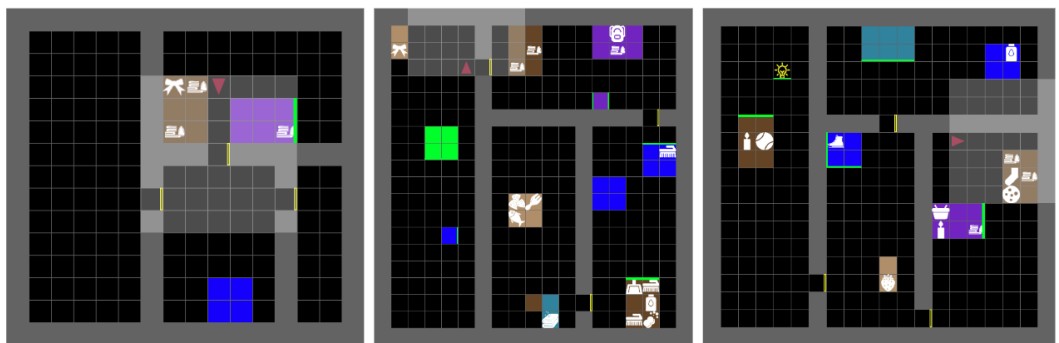

Figure B.1: Examples of the Visual Representation of the MARPLE Simulation Environment.

# B    MARPLE Household Simulator: Details

The MARPLE Household Simulator consists of two components: a multimodal simulator and a hierarchical agent planner.

## B.1    Simulator: A Multimodal GridWorld Environment

The simulator is built on top of Mini-BEHAVIOR [22], a GridWorld environment that is fast, simple, and easy-to-use. It supports procedural generation of diverse environments, symbolic states, and high-level agent actions, making it suitable for simulating realistic, long-horizon tasks.

Our simulator inherits several features from Mini-BEHAVIOR, including the standard $m \times n$ grid layout and asset library of furniture and object classes, action space, and state space. The asset library statistics are in Table B.1.

Our simulator further extends Mini-BEHAVIOR to support multimodal stimuli as follows: **Visual.** The visual representation of our environment is a $m \times n$ grid of cells. We inherit Mini-BEHAVIOR's visualization of agents, objects, and furniture, which are represented as triangles, icons, and colored backgrounds, respectively. Each cell can contain an object and a furniture, and the furniture states are indicated by green borders along the cell edges.

Each environment state has a corresponding array and a scene graph representation. An $m \times n$ environment has a $m \times n \times 8$ array representation. The 8 channels indicate the room type, furniture type, furniture states, object types, object states, object ids, agent position, and agent direction at each cell.

Meanwhile, the scene graph representation is a standard scene graph with a set of nodes and directed edges. The nodes represent entities, and the directed edges represent physical relations between entities, such as object-object relations and object-room relations.

**Language.** Our simulator supports two kinds of language descriptions that can be generated by an agent: intent and testimony. An agent's intent describes an action that they are about to perform, e.g. *"I am going to open the closet in the Bedroom."* An agent's testimony provides information on previous state changes in the environment that it observed, e.g. *The clothes in the closet in the Bedroom were picked up.* The language descriptions are generated from templates which takes in the action and relevant room and objects.

**Audio.** To simulate the sounds produced by agent actions, we incorporate realistic audio recordings and define an action-audio mapping. The audio files are obtained from `https://freesound.org`, and they are clipped to be 1 second long.

Table B.1:  MARPLE Household Simulator Elements Type Statistics.

| Environment Elements | | | Behavior Elements | | Engine Elements |
|---|---|---|---|---|---|
| Room Types | Furniture Types | Object Types | Mission Types | Action Types | State Types |
| 6 | 22 | 82 | 10 | 10 | 18 |

```
{"Grid":{
    "auto": {
        "max_num_agent": 5,
        "room_split_dirs": ["vert", "horz"],
        "min_room_dim": 5,"max_num_room": 4}, (universal settings)
        "width": 15, "height": 15,
    "agents": {
        "num": 1,
        "Initial":[{
            "name": "A","gender": "male","pos": [13, 13],"dir": null,
            "color": "red","step_size": 1,
            "mission_preference_initial": {"get_night_snack": 1},
            "cur_mission": null,"cur_subgoal": null,"carrying": null}]}
        "rooms": {
            "num": 2,
            "Initial":[
                {"type": "Bedroom", "top": [1, 1], "size": [9, 13],
                    "furnitures":{"num": 2,"initial":[
                        {"type": "bed","state": null,"pos": [1, 1],
                            "objs": {"num": 1,"initial":[{
                                "type": "remote","state": null,"pos": null}]}},
                        {"type": "table","state": {"dustyable": 1},"pos": [6, 6],
                            "objs": {"num": 0,"initial": []}}]}},
                {"type": "Kitchen","top": [11, 1],"size": [3, 13],
                    "furnitures":{"num": 2,"initial":[
                        {"type": "light","state": {"toggleable": 1}, "pos":[12,3],
                            "objs": {"num": 0, "initial": []}},
                        {"type":"electric-refrigerator", "state": {"openable": 1}, "pos":[12,10],
                            "objs": {"num": 1, "initial":[{
                                "type": "sandwich", "state":null,"pos":null}]}}
            ]
        }
    }
}
```

Figure B.2: Example of a simple configuration json file for the mission: `get night snack`.

## B.2 Planner: A Hierarchical Planner for Agent Behavior Generation

The hierarchical planner generates agent trajectories given its `mission preferences`, a distribution over all possible missions. It consists of 3 components: a high-, mid-, and low-level planner.

**High-Level Planner**. The high-level planner first samples a mission according to the agent's mission preferences. If the current mission becomes infeasible at any point, the current mission terminates, and the high-level planner resamples a new mission.

**Mid-Level Planner**. Given a mission, the mid-level planner is a Finite State Machine that determines the next subgoal to accomplish. It is given the sequence of subgoals to accomplish, and it finds the first one in the sequence that has not been executed yet. If the first unaccomplished subgoal is not feasible, (e.g. there is no light in the Kitchen), the current mission terminates.

**Low Level Planner**. The low-level planner decomposes a subgoal into a sequence of agent actions to accomplish the subgoal, using the A-star algorithm. It generates the shortest path to navigate to the target object, positions the agent, and performs the specified action. The simulator then propagates the environment state based on these actions. When a feasible trajectory is found, the trajectory is saved; otherwise, the current mission terminates.

## B.3 Usage: Ensuring diversity and complexity

Inference scenarios are procedurally generated according to a configuration file, as shown in Figure B.2. This file specifies initial conditions such as objects, states, and positions. Optional constraints include environment size, number of additional rooms, furniture, objects, and their positions.

The environment is first instantiated with the specified elements, and the additional ones are randomly selected from the asset library. They are placed randomly throughout the environment, resulting in diverse environment instances. The planner then generates agent trajectories within the environment.

To ensure complexity, the environment size, number of objects, number of rooms can all be scaled as needed. An $m \times n$ environment has a $m \times n \times 8$ state representation, causing the state space to grow exponentially with the array size.

## C MARPLE Dataset

### C.1 Dataset Details

**Dataset description.** We provide a dataset description in a datasheet: `https://github.com/marple-benchmark/marple/blob/main/datasheet.md`.

**Link and license**. The dataset is uploaded for public download at `https://drive.google.com/drive/folders/1zXsErNVOMYjBMWzTnmZS4e4aIljWlRce?usp=sharing`. It will be released under the CC-BY-4.0 license.

**Author statement.** The authors bear all responsibility in case of violation of rights. All dataset trajectories were collected by the authors and we are releasing the dataset under CC-BY-4.0.

**Format.** The data is uploaded in a simple zip format, with a zip file for each inference scenario in each train and test dataset. Upon decompressing the archive, a directory is provided for each instance that contains two subdirectories, one per agent. These are named with the agent's mission, and they contain files for the array and scene graph representations of each step in the trajectory, labelled by the timestep.

### C.2 Data Generation

For each inference scenario, we provide training and testing datasets. Each testing dataset contains 500 paired trajectories, instantiated in 10 diverse, procedurally generated rooms. We provide two types of training sets, each containing 5000 paired trajectories. For one type, 500 trajectories are generated in each of the 10 testing environments. For the second, 5000 environments are procedurally generated with 1 trajectory each. The configuration files used to generate all of the data are provided in our codebase.

# D   Computational Resources and Experiment Details

## D.1   MARPLE Simulator: Computational Resources

Our simulator operates at 600 frames per second (FPS) and requires only 1 frame for a primitive action. We run our experiments on the Stanford SC computational cluster with 1 NVIDIA TITAN RTX GPU and 8 CPU per job. With these resources, each inference trial takes 1.5 hours. The speed and efficiency of our simulator allows researchers to effectively evaluate their methods and focus on solving high-level, long-horizon inference challenges.

In contrast, a realistic physical simulator such as BEHAVIOR [26] runs at 60 FPS and requires 100 frames to perform a primitive action, making larger-scale experiments impractical. Such detailed physics simulation is also unnecessary for our inference setup, which focuses on understanding high-level agent behavior rather than physical interactions or photorealistic rendering.

## D.2   Experiment Resource Requirements

We ran experiments on the Stanford SC computational cluster with 1 NVIDIA TITAN RTX GPU, 8 CPU, and 30 GB RAM for each job. With these resources, each trial for a mental-simulation baseline took 1.5 hours to run. Each trial for `GPT-4` took 1 minute to run and required 32 API calls, resulting in a cost of $11 * 8 * \$0.50 = \$44.00$ per trial.

We evaluate each baseline on 50 trials. Each mental-simulation baseline took 75 hours total (jobs were submitted in parallel), and we evaluate on 4 variants of the simulation baseline for a total of 300 hours. For GPT-4, the 50 trials took 1 hour and cost \$2200. For humans, it took roughly 3 hours to complete the set of 50 trials.

## D.3   Statistical Significance

We choose to evaluate on 50 trials. This provides a good balance between statistical power and computational resources, as performing inference for a single trial is resource-intensive.

We plot the inference accuracy across the 50 trials with 95% CI, as shown in Figure 4 and Figure 6. The error bars in Figure 4 and Figure 6 are calculated using the standard formula: $CI = \bar{x} \pm \frac{\sigma}{\sqrt{n}}$, where $\bar{x}$ is inference accuracy, $\sigma$ is standard deviation, and $n = 50$ is the number of trials. Our figures indicate that 50 trials is sufficient, as the error bar is small enough to draw meaningful conclusions.

**Algorithm E.1** Simulation with Monte Carlo Sampling and Learned Agent Models

---

1: **Input:** Observations of both agents $o_\tau^A, o_\tau^B$
2: **Output:** $P(A), P(B)$ that $A$ or $B$ caused $s_T$
3: Initialize $count \leftarrow 0$
4: **for** $i \leftarrow 0$ to $m - 1$ **do**
5:    **for** $t \leftarrow \tau$ to $T$ **do**
6:       Sample $a_t^A$ according to $P(a|\pi^A, o_t^A)$
7:       Pass $a_t^A$ to the simulator, obtain $s_{t+1}^A, o_{t+1}^A$
8:       **if** $s_{t+1}^A = s_T$ **then**
9:          $count \leftarrow count + 1$
10:          **break**
11:       **end if**
12:    **end for**
13: **end for**
14: $P(s_T|\pi^A, o_{0:\tau}^A) \leftarrow count/m$
15: Repeat 3-14 for agent $B$ to get $P(s_T|\pi^B, o_{0:\tau}^B)$
16: Normalize using Equation (1) to get
   $P(A), P(B) =$
   $\text{softmax}(P(s_T|\pi^A, o_{0:\tau}^A), P(s_T|\pi^B, o_{0:\tau}^B))$

---

# E   Simulation with Learned Agent Models: Details

## E.1   Algorithm

Algorithm E.1 is used to perform simulation with Monte Carlo sampling and learned agent models.

## E.2   Implementation Details

**Agent Model Architectures.** We have four variations of our agent policy models: vision-only, audio-augmented, language-conditioned, and audio-augmented language-conditioned.

The vision-only and audio-augmented policy models are implemented with a Vision Transformer (ViT) as an encoder with a multi-layer perceptron (MLP) to predict the agent actions. After experimenting with different model and layer sizes, we use a ViT encoder with an image size of $20 \times 20$, patch size of $1 \times 1$, depth of 15, embedding dimension of 1024, 8 channels, and 16 heads and a 4-layer MLP with intermediate ReLU layers.

The language-conditioned and audio-augmented language-conditioned policy models are transformer-based with a ViT encoder and 4 decoders for the object, furniture, room, and action. Each decoder is a 2-layer MLP with an intermediate ReLU layer. After experimentation, we use a ViT encoder with an image size of $20 \times 20$, patch size of $1 \times 1$, depth of 15, embedding dimension of 1024, 8 channels, and 16 heads. Each decoder has an input dimension of 256, hidden dimension of 256, position embedding dimension of 64, depth of 8, dropout of 0.1, and gelu activation.

**Agent Model Training Data.** We learn agent models for all 10 of the provided missions. We train our agent models on two types of agent behavior datasets, as described in Appendix C.

**Agent Model Training Details.** We perform sweeps for hyperparameter tuning using WandB. Ultimately, we train our low-level policy models using a batch size of 64 and a learning rate of 1e-4, optimized with the Adam optimizer. The models are trained for 20 epochs, and this includes a gradual warmup scheduler with a multiplier of 1 and a warmup period of 4 epochs, followed by a cosine annealing learning rate scheduler over the remaining epochs. Additionally, we employ gradient accumulation to enhance the training efficiency and stability.

# F  Ablations of Simulation with Learned Agent Models

We provide extensive ablation of our simulation baselines, and we explore the effect of each modality (vision, audio, and language) on performance. These demonstrate that the vision-only baseline performs the worst, and the addition of audio and language are both beneficial. While language seems more valuable than audio in inference, the baseline using all 3 modalities consistently outperforms the others. This suggests that audio and language provide useful, distinct information in inference.

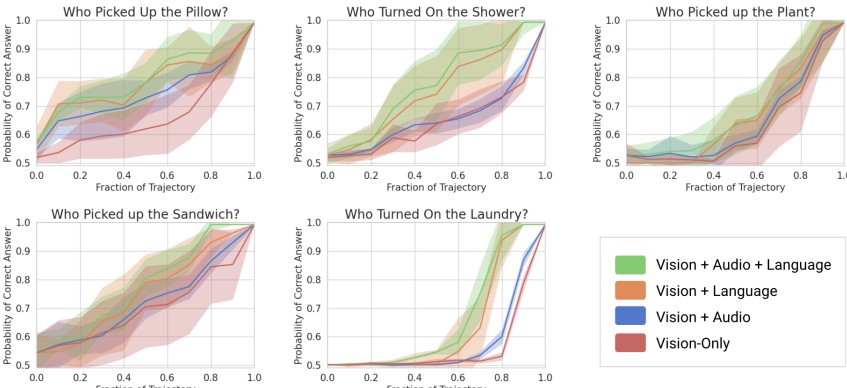

Figure F.1: Performance of each variant of the simulation baseline on all 5 inference scenarios. These baselines are tested in-distribution, on the same environments seen in training. The vision-only baseline performs the worst. While language seems more useful than audio, the baseline with all 3 modalities consistently outperforms the others. This suggests that both audio and language provide useful, distinct information.

# G  Prompts for GPT-4

We provide the prompt templates for GPT-4:

---

**Prompt illustration for generating completions**

```
Instructions:
Take a deep breath.  Your task is to analyze and determine which agent (target agent,
other agent) is more likely to have performed specific actions leading to the final
state of the environment.

Remember, the states you are analyzing are select snapshots from a larger sequence.
If the agents have gone through e.g., 100 states, you might only be seeing a fraction
of these (like every 10th state for each agent), which means critical movements and
decisions may have occurred in the unseen states.

Initial State of Target Agent:  [state here]

Current State of Target Agent:  [state here]

Initial State of Other Agent:  [state here]

Current State of Other Agent:  [state here]

Final State:  [state here]

Your analysis should consider how the changes and progression from the initial to
the current state for each agent might indicate their likely actions in the final
state.  Reflect on the sequence of events and decisions made by each agent.  Based
on analyzing the changes between the initial and current states, and the final state,
you must answer the following question about the final state:

Question:  [inference question here]

Answer Options:
Provide an integer between 0 - 100 (where 0 = definitely target agent and 100 =
definitely other agent)

Strictly follow this response format:

Reasoning:  [detailed 'Let's think step-by-step...'  reasoning]
Answer:  [answer as an integer between 0 and 100 here]
```

---

Figure G.1: Prompt template (simplified) for generating completions with GPT-4.

# H  Additional Results of Open-Source LLMs

We present additional results evaluating top state-of-the-art open-source LLMs (Llama-3.1-8B-Instruct and Qwen2-7B-Instruct) on our benchmark. We choose these models due to their large context length, as our prompt is over 11,000 tokens.

Both LLMs struggle to perform the inference task. Llama-3.1's performance is lower than but consistent with GPT-4's. For scenarios where GPT-4 does converge, Llama-3.1 does not necessarily converge, but it shows an increase in inference accuracy as the trajectory progresses, indicating some signal. For scenarios where GPT-4 does not converge ("Who turned on the shower" and "Who turned on the laundry"), Llama-3.1's inference accuracy does not improve with later evidence. We find that Llama-3.1 often reasons correctly about the state changes between timesteps, but it does not arrive at the correct conclusion. Meanwhile, Qwen2's inference accuracy does not increase as the trajectory progresses and struggles to reason accurately about the state changes.

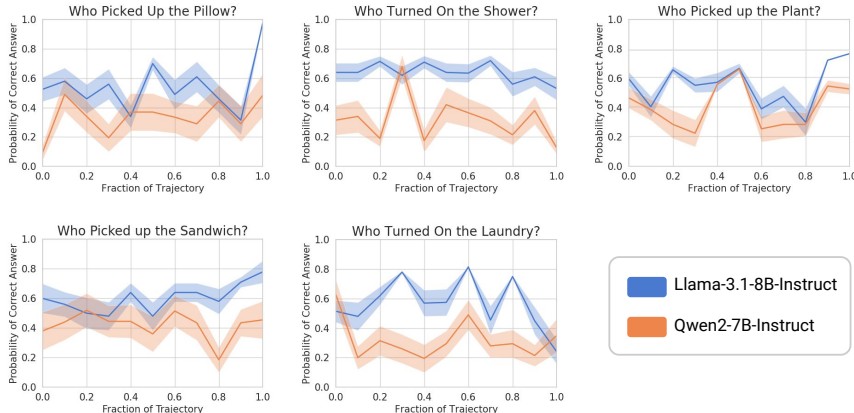

Figure H.1: Performance of state-of-the-art open-source LLMs on all 5 inference scenarios.

# I  Additional Results of `GPT-4` with In-Context Learning

We conduct additional experiments using `GPT-4` with in-context learning (ICL). We evaluate on the two scenarios where `GPT-4` failed to converge with zero-shot prompting: "Who turned on the shower" and "Who turned on the laundry."

As shown in Figure I.1, `GPT-4`'s performance improves with ICL — it fluctuates less and ends with a higher accuracy than the zero-shot baseline. However, it still fails to converge. In Appendix J, we provide examples of `GPT-4` step-by-step reasoning to analyze this failure mode.

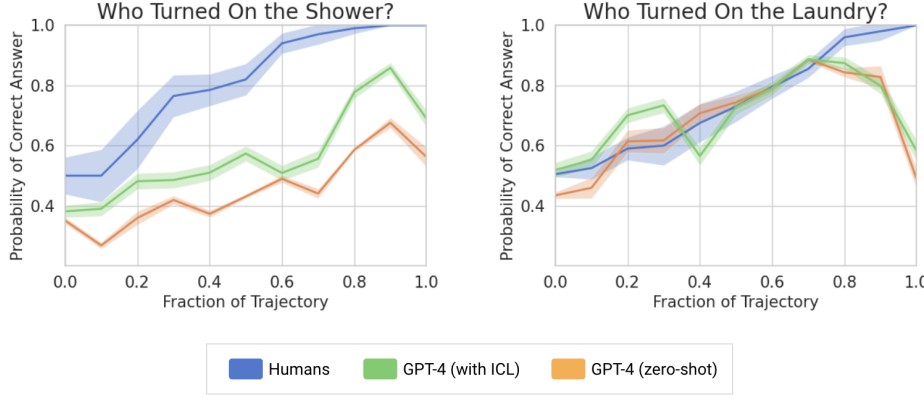

Figure I.1: Performance of GPT-4 baselines on two selected inference scenarios with the largest gap between human and GPT-4 performance.

# J   Analysis of `GPT-4` Reasoning

We provide examples of `GPT-4`'s incorrect reasoning in the "Who turned on the laundry?" scenario, where Agent A is performing `do laundry` and Agent B is performing `change outfit`. The correct answer is Agent A. In the input state for these examples, Agent A is stationary and turning on the laundry while Agent B is moving and carrying clothes. The key evidence is reflected in the change in environment state for Agent A: the laundry's `ToggledOn` state changes from `False` to `True`.

With both zero-shot prompting and in-context learning, we notice that `GPT-4` tends to rely on agent state information and less on environment state information. This causes `GPT-4` to make the wrong inference in this case.

---

**Sample Reasoning from GPT-4 (zero-shot)**

```
Let's think step-by-step and analyze the changes between the initial and current
states for each agent.  For the Agent A, the initial and current states are
identical.  The agent's position, direction, and carried items have not changed.
However, we can notice a change in the state of the laundry in the Bathroom.
Initially, the laundry was not toggleable, but in the current state, it became
toggleable.  However, there is no evidence that the Agent A has moved or interacted
with the environment.

On the other hand, the Agent B has shown some activity.  Initially, the Agent B was
carrying a piece of clothes, and the agent's position was [9, 5].  In the current
state, the Agent B has moved one step north to [9, 6], but the agent is still
carrying the same piece of clothes.  There are no changes in the state of the laundry
in the Bathroom for the Agent B.

In the final state, the laundry in the Bathroom is toggleable, which is the same as
in the current state of the Agent A. However, there is no clear indication that the
Agent A has moved or interacted with the laundry.  On the other hand, the Agent B
has shown activity by moving, but there is no direct evidence that the Agent B has
interacted with the laundry.

Given these observations, it's difficult to definitively say which agent is more
likely to have turned on the laundry.  However, since the Agent A's environment shows
a change in the state of the laundry and the Agent B has shown activity (despite no
direct interaction with the laundry), we can lean towards the Agent B.
```

---

**Sample Reasoning from GPT-4 (in-context learning)**

```
Reasoning:  Agent A started at position [13, 5] and remained at the same position in
the current state.  The direction of Agent A was 3 in the initial state and remained
3 in the current state.  This indicates that Agent A has not moved at all from the
initial state to the current state.

Agent B started at position [15, 9] and moved to position [4, 12] in the current
state.  The direction of Agent B was 1 in the initial state and changed to 3 in the
current state.  This indicates that Agent B has moved a considerable distance and
changed direction.

The final state shows an agent at position [13, 5] with direction 3.  This is exactly
the same position and direction as Agent A in both the initial and current states.
The final state also shows that the laundry is now turned on.  Given that Agent A
has not moved at all, it is highly unlikely that Agent A could have turned on the
laundry.  On the other hand, Agent B has shown movement and change in direction,
indicating that it is capable of performing actions.

Therefore, it is more likely that Agent B is the one who turned on the laundry.

Given that Agent A hasn't moved at all and Agent B has shown the capability of
interacting with objects (by picking up the 'clothes'), it's more likely that Agent
B is the one who moved the 'clothes' to the 'laundry' and turned it on, despite the
final position and direction matching with Agent A.
```

# K   Details on Human Experiments

We conduct experiments with 2 human experts. Each participant was provided with a habituation phase, in which they were familiarized with MARPLE domain knowledge, the inference setup, and a few examples of the agent trajectories beforehand. Each human participated in 50 inference trials which took around 3 hours.

For each trial, we show participants two agent trajectories, shown side-by-side with labels "Agent A" and "Agent B". They start from the initial step and move to the next timestep at their own pace, until they reach the end. This allows them to incrementally build an understanding of the agent trajectories and compare agent behaviors within the scenario. A diagrammatic illustration of the human study is shown in Figure K.1.

As they view the trajectories, we ask them to answer the inference question, e.g. "Which agent is more likely to have turned on the laundry?", at 11 evenly spaced timesteps, consistent with the mental-simulation and LLM baselines. The participants indicate their prediction using a scale from 0 to 100, with 0 being "definitely agent A" and 100 being "definitely agent B".

**Which agent is more likely to have turned on the laundry?**

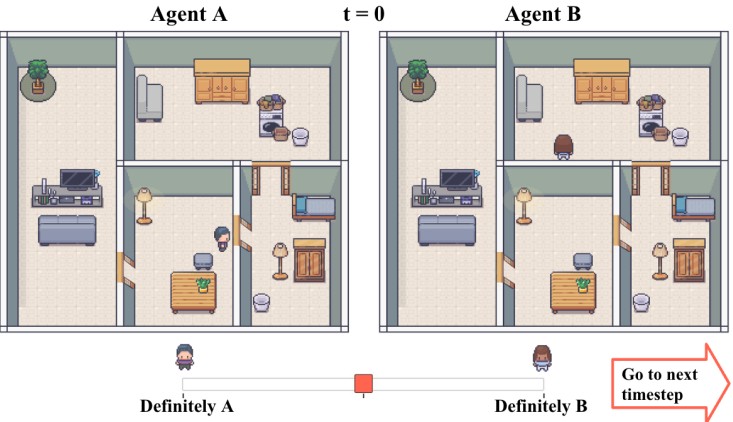

Figure K.1: Diagrammatic illustration of the human study for MARPLE. Participants saw Gridworld versions of the scenes. They started with initial scene, clicked the arrow sign to move to the next step, and then responded to the inference question by dragging the slider.

