# OpenReview forum: "MARPLE: A Benchmark for Long-Horizon Inference"
_NeurIPS.cc/2024/Datasets_and_Benchmarks_Track — NeurIPS 2024 Track Datasets and Benchmarks Poster_

### Official Review · Reviewer_shNX · 2024-06-22
**Promising Concept with Room for Improvement**

**Rating:** 7
**Confidence:** 3
**Clarity:** Yes

**Review:**

Quality: The paper presents a well-constructed benchmark with detailed methodologies for generating environments and evaluating inference tasks. However, the experimental setup, particularly the choice of models and their configurations, requires more rigorous validation to ensure the findings are robust and generalizable.

Clarity: The paper is generally well-written and structured, but certain sections, such as the experimental results, would benefit from clearer explanations and more comprehensive statistical analysis.

Originality: The benchmark addresses a unique and underexplored area in AI inference, contributing valuable insights into the limitations of current models in long-horizon reasoning tasks.

Significance: The benchmark has significant potential to advance research in AI inference, though its immediate impact might be limited by the need for further validation and refinement of the experimental approaches.

Pros:

Innovative Benchmark: MARPLE fills a gap in evaluating long-horizon, multi-modal inference, an area not sufficiently covered by existing benchmarks.
Comprehensive Approach: The combination of vision, language, and audio stimuli in a procedurally generated environment provides a robust testbed for AI models.
Human Baseline: Including human performance as a benchmark provides a valuable reference point for evaluating AI model capabilities.
Cons:

Experimental Validation: The choice of baseline models and their configurations (e.g., Monte Carlo simulation, GPT-4) needs more rigorous justification and exploration. The performance differences could be better explained with additional ablation studies and sensitivity analyses.
Statistical Rigor: The analysis lacks detailed statistical testing to confirm the significance of the results. Confidence intervals are mentioned, but deeper statistical comparisons are necessary.
Generalizability: The benchmark's scenarios, while diverse, might not cover the full spectrum of real-world complexities. More extensive testing in varied environments is needed to validate the benchmark's comprehensiveness.
Model Limitations: The paper does not sufficiently address the limitations of the evaluated models, particularly the reasons behind GPT-4's underperformance in specific tasks. More detailed error analysis would be beneficial.

**Strengths:**

- Tackling an important but underexplored problem of long-horizon, multimodal reasoning that is crucial for advancing AI capabilities towards human-level intelligence.
- Proposing a new simulation environment and procedural generation approach that enables systematic data collection and research into these challenging reasoning tasks.
- Providing a thorough empirical analysis of several state-of-the-art baseline methods compared against human performance, surfacing clear limitations of current models.
- Careful experimental design and thoughtful articulation of the reasoning skills required to succeed on the proposed MARPLE benchmark.
- Potential impact by stimulating further research into multimodal integration and long-range inference, which could have broad applicability across AI tasks.

**Additional Feedback:**

MARPLE represents a good work in evaluating AI models' capabilities in complex inference tasks, and with the suggested improvements, it has the potential to become a cornerstone benchmark in the field.

**Correctness:**

The claims made in the submission are generally well-supported by the provided experiments and analysis. The benchmark is constructed in a sound manner, with clear definitions of the inference tasks and thorough procedural generation of environments and agent behaviors.

- Appropriate Methods: The use of Monte Carlo Tree Search and GPT-4 as baselines is appropriate for the benchmark's objectives. However, the evaluation could be strengthened by including more diverse models and additional ablation studies to better understand the impact of different modalities and configurations.
- Experimental Design: The experimental design is robust, but the statistical analysis could be more detailed. Including more rigorous statistical tests and error analysis would enhance the reliability of the findings.

**Documentation:**

The documentation for MARPLE is comprehensive, with sufficient detail on the procedural generation of environments, agent behaviors, and the multi-modal stimuli used. The authors also provide clear instructions for reproducing the experiments, including code, datasets, and pre-trained models.

Areas for Enhancement:

- Data Collection and Organization: While the procedural generation is well-documented, additional information on how the generated data is organized and managed would be beneficial. Details on data storage, access, and version control would support reproducibility and maintenance.

**Ethics:**

There are no immediate ethical concerns with the submission that warrant a specialized ethics review. The authors have ensured that their work conforms to ethical guidelines and does not involve sensitive data or human subjects in a way that raises significant concerns.

Potential Ethical Considerations:

- Bias and Fairness: Ensuring that the benchmark scenarios are diverse and representative of various real-world situations can help mitigate biases. The authors should consider including a discussion on how they address potential biases in the procedural generation process.
- Privacy and Consent: While the current setup does not involve personal data, future expansions, especially those incorporating more realistic simulations or human data, should consider privacy and consent issues.

**Limitations:**

The authors have made a good effort in addressing the limitations of their work, particularly in Section 7. They acknowledge the constraints of their simulation environment, such as the GridWorld's lack of physical realism, which may limit its suitability for low-level reasoning tasks. They also recognize the limitations of their language and audio stimuli, generated from a defined library, and the current two-agent setup, which can be extended to support multiple agents.

Constructive Suggestions:

- Expand Realism: Future iterations of the benchmark could benefit from incorporating more physically realistic simulations. This would enhance the applicability of the benchmark to real-world scenarios and potentially improve the robustness of evaluated models.
- Diverse Language and Audio Stimuli: Enhancing the diversity and naturalness of language and audio stimuli by including free-form natural language descriptions and realistic audio renderings would provide a more comprehensive testbed.
- Multi-Agent Scenarios: Exploring multi-agent setups beyond two agents could introduce additional complexity and realism, offering deeper insights into models' inference capabilities in more dynamic environments.

**Opportunities For Improvement:**

1) Increasing the complexity, realism and diversity of the simulated environments and multimodal data beyond the current gridworld/template-based approach.
2) Expanding the types of inference queries and tasks beyond just the "whodunit" event attribution to support more general state/event reconstruction.
3) Developing new end-to-end learnable models specifically designed for these long-horizon multimodal reasoning tasks, rather than just applying imitation learning and large language models.
4) Deeper dive into analyzing the key factors and failure modes that make these reasoning problems difficult through more granular probing studies.
5) Providing prescriptive insights on the most promising future research directions based on the limitations uncovered in current models.
Considering broader societal impacts, risks, or ethical concerns that could arise from improving AI models' multimodal reasoning capabilities in complex environments.

**Relation To Prior Work:**

The paper provides a clear discussion on how MARPLE differs from previous benchmarks, particularly in its focus on long-horizon, multi-modal inference tasks. The comparison with other benchmarks in Table 1 effectively highlights MARPLE's unique contributions


Suggestions for Improvement:

- Deeper Analysis: A deeper analysis of the specific gaps in prior work that MARPLE addresses would strengthen the discussion. For example, detailing why existing benchmarks are insufficient for long-horizon inference could provide more context.
- Broader Context: Discussing how MARPLE can be integrated with or complement existing benchmarks would offer a broader perspective on its potential impact on the field.

**Summary And Contributions:**

This paper introduces MARPLE, a novel benchmark designed to evaluate AI models' ability to perform long-horizon inference using multi-modal evidence. The benchmark simulates "whodunit" scenarios in household environments where AI models must identify which agent caused a specific change. The paper details the benchmark's design, its procedural generation of environments and agent behaviors, and the evaluation of AI models against human performance. The primary contributions include:

- A comprehensive benchmark for long-horizon, multi-modal inference tasks.
- A procedural environment generator for diverse household scenarios.
- Benchmarking results comparing traditional Monte Carlo simulation, GPT-4, and human performance.

---

> ### Author Rebuttal · Authors · 2024-08-16
>
> Thank you for your positive evaluation and for your valuable feedback regarding more extensive ablation studies, testing in varied environments, and opportunities for future improvement. Please feel free to let us know if further clarification is needed, and we would be happy to provide it.
>
> **Experimental Validation**
>
> Following your suggestion, we provide more extensive ablation results (attached PDF, Figure 2) of our Monte Carlo baselines, and we explore the effect of each modality (vision, audio, and language) on performance. These demonstrate that the vision-only baseline performs the worst, and the addition of audio and language are both beneficial. While language seems more valuable than audio in inference, the baseline using all 3 modalities consistently outperforms the others. This suggests that audio and language provide useful, distinct information in inference.
>
> **Generalization Experiments**
>
> The environments used in our experiments contain diversity in terms of the object types, object placement, and room layouts. We provide new results (attached PDF, Figures 2 and 3) to highlight that our current level of diversity presents a challenge for current models. In Figure 1, the baselines are trained in the same environments that they are tested on, while in Figure 2, they are trained on procedurally generated environments and tested on new, unseen ones. Performance drops significantly when tested on unseen environments. Even the baseline using all 3 modalities is unable to make the correct inference with a probability of 0.8 until they see around 80% of the trajectory, demonstrating the challenge of our benchmark's diversity.
>
> **Additional GPT-4 Analysis**
>
> We agree that it would be valuable to study GPT-4’s performance. In [Section J](https://openreview.net/attachment?id=nAFBHoMpQs&name=supplementary_material#page=8.76) of our supplementary material, we provide analysis of GPT-4’s failure cases of a task where it underperforms, and we plan to add more examples to the final manuscript.
>
> **Opportunities for Future Improvement**
>
> Thank you for pointing out opportunities to increase the complexity, realism and diversity of the simulated environments, expand the types of inference queries and tasks beyond "whodunit", develop new end-to-end learnable models, and conduct further studies on the AI models' multimodal reasoning capabilities. These are all exciting directions of future research that we hope to address and that our benchmark hopes to support.
>
> **Data Documentation**
>
> Thank you for the suggestion. We have updated the Data section of our Github README to include more details on how the data is generated and organized.

---

> > ### Author Response · Authors · 2024-08-26
> > **Happy to answer any further questions**
> >
> > Dear Reviewer shNX,
> >
> > Thank you for your detailed review our submission. We have put great effort into addressing your questions during the rebuttal period, specifically providing additional experiments on our Monte Carlo baselines, generalization experiments, and improved data documentation. We are excited to discuss with you and answer any further questions. Thank you again for taking the time to read our work, and we look forward to your feedback!
> >
> > Thank you,
> >
> > Authors

---

> > ### Author Response · Authors · 2024-08-29
> >
> > Dear Reviewer shNX,
> >
> > Thank you again for your valuable feedback on our submission. We have posted our clarification and response, and we would be happy to hear your updated thoughts. We look forward to further discussions and are happy to answer any additional questions!
> >
> > Thank you,
> >
> > Authors

---

### Official Review · Reviewer_AsPM · 2024-07-25
**Interesting simulator and benchmark, storyline may need adjustments**

**Rating:** 8
**Confidence:** 4
**Clarity:** The paper is clear and well-written.

**Review:**

The paper is generally very nice and introduces a very interesting long-horizon inference problem. Please see the "Strength" for what I like about this paper. Here I would like to talk about some questions and comments to start a discussion during the rebuttal phase:

(1) The storyline of "whodunit" sounds initially very fascinating. However, after thinking about it, I do not think it is a very proper storyline. This paper mostly solves the problem of "who will do it" -- based on the agents' past behavior to infer their future plans, i.e., $P(S_T|\pi^i, o_{0 : \tau})$. "Whodunit", however, if using the same formulation as the paper, should be written as $P(S_T|\pi^i, o_{\tau : T})$. In other words, "Whodunit" is not an inference problem but a memory problem, as we trace back step by step and the answer is trivial when $\tau=0$. So I believe the paper is solving a different problem rather than a whodunit.

(2) I suggest adding more information about how to configure and run the simulator. I looked through the supplementary materials and the  GitHub link but did not find much. I expect such simulators can be used for other types of inference study, such as complex event detection in long horizons (e.g., detect the emergence of a meaningful sequence of events that span across a large space and a long time).

(3) I am not sure if "language" is a good modality to be included in this work. When I observe these experiments, it seems like the language modality is actually having the agents speak out their intentions, which leaks information about their future plans. On the other hand, vision and audio are stateless observations that won't "leak the future".

(4) In terms of LLM, only GPT-4 is tested. Are other types of LLMs showing a similar type of inference failure? I understand this paper is not really about benchmarking LLM, but more experiments with other variants of LLM may provide more insight and a more decisive conclusion.

**Strengths:**

(1) Nice simulator which requires a lot of effort
(2) Comprehensive details provided in the supplementary material
(3) Good potential for machine cognism research
(4) Clear problem formulation

**Additional Feedback:**

Softmax equation on Page 2 is missing parenthesis on the denominator? It should be exp(A)/(( exp(A))+( exp(B)))

**Correctness:**

The claim is generally correct. I have a little disagreement with the addition of language and with the storyline formulation.

**Documentation:**

Proper documentation is provided.

**Ethics:**

The paper does not collect human data directly but has a human-involved user study. I would like to suggest that ACs call for ethic reviews at your discretion. To me, there is no privacy leaking or potential harm to the study participants.

**Limitations:**

Limitations are discussed properly in Section 7.

**Opportunities For Improvement:**

See the "Review" section for details.

**Relation To Prior Work:**

Related work is properly discussed.

**Summary And Contributions:**

The paper describes a simulator, a dataset, and associated benchmarking experiments. They leverage a simulator and an activity generation algorithm to produce long event sequences and ask different models to predict which one of the agents in the simulator is going to perform a particular action in the future. Monte Carlo Tree Search, LLM, and humans are being tested, where human significantly outperforms.

---

> ### Author Rebuttal · Authors · 2024-08-16
>
> Thank you for your positive evaluation of our work and for your valuable feedback, which has helped us to clarify our storyline, and improve our simulator documentation. We also provide new results of open-source LLMs in our attached PDF (Figure 1). Please feel free to let us know if further clarification is needed, and we would be happy to provide it.
>
> **Whodunit Storyline**
>
> Thank you for your feedback on our storyline. Our benchmark is motivated by ‘whodunit’ problems, and we formulate the problem we address in our paper in a way that serves as a stepping stone to solving such problems. Solving a ‘whodunit’ problem requires tracing back the evidence and performing a mental simulation of what will happen. So, having a good forward simulator of how a person acts is necessary for later performing the inference of who did it. We will update the introduction of our paper to further clarify the motivation and problem formulation.
>
> **Simulator Documentation**
>
> Following your suggestion, we have added a section in our [README](https://github.com/marple-benchmark/marple/blob/main/README.md#simulator-configuration) on how to configure and run our simulator, along with a demo script.
>
> **Clarification on Language Modality**
>
> The language modality only reveals future information about the subgoal that the agent is intending to perform next, so it does not reveal their mission until the ultimate subgoal. The challenge is to effectively leverage knowledge about how these intentions are related to the final state. We carefully constructed scenarios where the language modality helps to varying degrees.
>
> For example, in the scenario “Who picked up the snack?”, the language evidence reveals early on that agent A intends to “open the refrigerator” while agent B intends to “pick up the towel from the closet.” From this, a strong inference model should be able to reason that agent A is more likely to pick up the snack.
>
> On the other hand, both agents share many subgoals in the scenario “Who toggled on the laundry”. Agent A performs: pickup clothes from the bed, open the laundry, drop clothes, close laundry, toggle-on laundry, while Agent B performs: open closet, pickup clothes from closet, close closet, open laundry, drop clothes, close laundry. In this case, the language evidence reveals nuanced information and only helps distinguish the agents’ missions at the end.
>
> **Additional LLM Results**
>
> Following your feedback, we present new results (attached PDF, Figure 1) evaluating top state-of-the-art open-source LLMs (Llama-3.1-8B-Instruct and Qwen2-7B-Instruct) on our benchmark. We choose these models due to their large context length, as our prompt is over 11,000 tokens.
>
> Both LLMs struggle to perform the inference task. Llama-3.1’s performance is lower than but consistent with GPT-4’s. For scenarios where GPT-4 does converge, Llama-3.1 does not necessarily converge, but it shows an increase in inference accuracy as the trajectory progresses, indicating some signal. For scenarios where GPT-4 does not converge (“Who turned on the shower” and “Who turned on the laundry”), Llama-3.1’s inference accuracy does not improve with later evidence. We find that Llama-3.1 often reasons correctly about the state changes between timesteps, but it does not arrive at the correct conclusion. Meanwhile, Qwen2’s inference accuracy does not increase as the trajectory progresses and struggles to reason accurately about the state changes. We hope that our benchmark supports future work aimed at enhancing LLMs’ inference abilities.
>
> **Softmax Equation**
>
> Thank you for pointing this out. We have updated this in our paper accordingly.

---

> > ### Comment · Reviewer_AsPM · 2024-08-27
> > **Increase my score from 7 to 8**
> >
> > I believe the authors have done their due diligence in addressing my comments. The provide both explanations, new experiments, and promised writing improvements. Since I like the problem formation and believe it will have potential in other domains such as complex event detection, I would increase the score by one point to advocate an acceptance.

---

> > > ### Author Response · Authors · 2024-08-29
> > > **Thank you**
> > >
> > > Dear Reviewer AsPM,
> > >
> > > Thank you again for your helpful comments and for reviewing our response! We are glad to hear that the concerns have been addressed and will incorporate all the contents in the rebuttal to our revised manuscript.
> > >
> > > Authors

---

> ### Author Response · Authors · 2024-08-26
> **Happy to answer any further questions**
>
> Dear Reviewer AsPM,
>
> Thank you for your detailed review our submission. We have put great effort into addressing your questions during the rebuttal period, specifically clarifying the Whodunit storyline, improving our simulator documentation, clarifying the language modality, and providing additional LLM results. We are excited to discuss with you and answer any further questions. Thank you again for taking the time to read our work, and we look forward to your feedback!
>
> Thank you,
>
> Authors

---

### Official Review · Reviewer_8BjY · 2024-07-25
**Review for MARPLE**

**Rating:** 6
**Confidence:** 4
**Correctness:** yes.
**Clarity:** Yes

**Review:**

This paper introduces a new benchmark that evaluates the performance of modal's ability in long-horizon tasks. The idea is novel and interesting and the data is also novel in its modality and task specification. However, there are still some concerns regarding the effectiveness of this data:

- Long-horizon problem is not well formulated. This paper directly relates the long-horizon context as the action step. However, there are some other scenarios such as the multi-hop reasoning, which are not discussed. Therefore, the long-horizon problem in this paper is not comprehensive and not well established.

- The number of evidence may not be relevant to the long-horizon information. This paper defined the number of evidence as the long-horizon information. However, some information in the trajectory may not lie in the long-horizon context. Ensuring this is important to validate the performance of the models.

- The human evaluation process is not well explained. The human evaluation is very important for finding the upper bound and potential challenges in identifying the long-horizon context, which is however not well explained.

**Strengths:**

- Novel ideas and multi-modality
- Experiments is comprehensively compared.

**Additional Feedback:**

NA

**Documentation:**

The human evaluation part is not clearly explained.

**Ethics:**

Human evaluation should be clearly discussed to ensure no ethical concerns.

**Limitations:**

- Ethical and fair comparison concerns. Since this paper conducted human evaluation, it is important to explain this process clearly

- Besides daily activities, some multi-hop reasoning tasks are also important to investigate the long-horizon tasks.

**Opportunities For Improvement:**

- The number of evidence may not be relevant to the long-horizon information

- The human evaluation process is not well established.

- Long-horizon problem is not well formulated.

**Relation To Prior Work:**

Yes

**Summary And Contributions:**

This paper builds a new benchmark dataset MARPLE that using Mini-BEHAVIOR simulator to collect a daily activities simulation data, which can evaluate the performance of the model in long-horizon.
The main contributions lie in the long-horizon tasks and the associated multi-modal data for evaluating the model's performance.
Moreover, this paper compares different baselines, including MCTS algorithm, LLM inference and human evaluation. There experiments comprehensively reflect the distinctions of different modalities.

---

> ### Author Rebuttal · Authors · 2024-08-16
>
> Thank you for taking the time to assess our manuscript and offer your valuable feedback, which has helped us to clarify our long-horizon problem formulation and human evaluation process. Please feel free to let us know if further clarification is needed, and we would be happy to provide it.
>
> **Long Horizon Problem Formulation**
>
> We agree that a large number of actions does not necessarily indicate that a scenario is long-horizon. However, we contend that in our setup, the number of actions is a strong measure of long-horizon context. The low-level actions in each agent trajectory are dependent on the mid-level subgoals, which in turn are dependent on the high-level mission, creating strong multi-step dependencies.
>
> For example, to complete the mission “do laundry”, an agent must perform low-level actions to: navigate to the bedroom closet → open the closet → pick up clothes from the closet → navigate to the bathroom → open the door → navigate to the laundry → open the laundry → drop the clothes in the laundry → turn on the laundry. Each subgoal relies on the completion of past ones and is necessary to complete future ones (e.g. the agent can only “pick up clothes from the closet” after “opening the closet,” and must have clothes before navigating to and dropping them in the laundry).
>
> Thus, our scenarios do require high-level reasoning over extended sequences of low-level actions. We will update our paper to further clarify this.
>
> **Number of Evidence Relevant to Long-Horizon Information**
>
> We understand the concern about the relevance of evidence to long-horizon information. In our setup, we use a hierarchical planner to generate sequences of low-level actions that directly accomplish each subgoal by navigating to the target location and performing the necessary action, avoiding unnecessary actions or random walks. Therefore, every action in the trajectory is relevant to accomplishing the mission, making the number of actions a valid measure of long-horizon information.
>
> For example, consider an agent that performs “do laundry” and turns on the laundry at timestep 100. Then, as it moves toward the laundry room at earlier timesteps, this evidence is relevant to the long-horizon information because direction of movement helps distinguish its long-term goal.
>
> **Human Evaluation Process**
>
> We provide additional details about how we conducted the human experiments in our supplementary material ([Section H](https://openreview.net/attachment?id=nAFBHoMpQs&name=supplementary_material#page=7.68)):
>
> “We conduct experiments with 2 human experts. Each participant was provided with a habituation phase, in which they were familiarized with MARPLE domain knowledge, the inference setup, and a few examples of the agent trajectories beforehand. Each human participated in 50 inference trials which took around 3 hours.
>
> For each trial, we show participants two agent trajectories, shown side-by-side with labels “Agent A" and “Agent B." They start from the initial step and move to the next timestep at their own pace, until they reach the end. This allows them to incrementally build an understanding of the agent trajectories and compare agent behaviors within the scenario. A diagrammatic illustration of the human study is shown in Figure H.1.
>
> As they view the trajectories, we ask them to answer the inference question, e.g. “Which agent is more likely to have turned on the laundry?”, at 11 evenly spaced timesteps, consistent with the mental-simulation and LLM baselines. The participants indicate their prediction using a scale from 0 to 100, with 0 being “definitely agent A" and 100 being “definitely agent B".”

---

> ### Author Response · Authors · 2024-08-26
> **Happy to answer any further questions**
>
> Dear Reviewer 8BjY,
>
> Thank you for your detailed review our submission. We have put great effort into addressing your questions during the rebuttal period, specifically clarifying our long horizon problem formulation, relevance of evidence, and human evaluation process. We are excited to discuss with you and answer any further questions. Thank you again for taking the time to read our work, and we look forward to your feedback!
>
> Thank you,
>
> Authors

---

> > ### Comment · Reviewer_8BjY · 2024-08-29
> > **Response to authors**
> >
> > Thanks for your explanation and clarification. I would like to raise my score to weak accept though with some concerns with the long-horizon.

---

> ### Author Response · Authors · 2024-08-29
> **Thank you**
>
> Dear Reviewer 8BjY,
>
> Thank you again for your helpful comments and for reviewing our response! We will incorporate all the contents in the rebuttal to our revised manuscript.
>
> Authors

---

### Official Review · Reviewer_fKzP · 2024-08-02
**Solid benchmark that can be improved with a bit more baseline comparisons.**

**Rating:** 7
**Confidence:** 4
**Correctness:** Yes.
**Clarity:** The paper is well written and easy to…

**Review:**

Strengths
- The problem is well motivated -- targeting inference focused tasks in a controllable setting is well needed to characterize how current AI systems still lags behind human efficiency in terms of performing inference
- Comprehensiveness of the environment design. The authors go full length in making the environment customizable. The different levels of planner for procedural behavior generation is also clear and well executed.

Weaknesses
- The baselines seem a bit slim; finetuning a weaker LLM based model could also tell us how easy LLMs can adapt to this type of distribution (either finetuning an 8B open weight model or finetuning through provided APIs)
- The audio element is a bit confusing to me. What motivates this type of audio input (or what does it add to the setup) since it's rather artificial mapping that reveals partial information about the language modality?

**Strengths:**

See review section.

**Additional Feedback:**

N/A

**Documentation:**

Yes.

**Ethics:**

No.

**Limitations:**

Yes.

**Opportunities For Improvement:**

See review section.

**Relation To Prior Work:**

Yes.

**Summary And Contributions:**

The authors propose a benchmark for the "whodunit" type inference problem. Building upon the Mini-BEHAVIOR simiulator, the authors extend the environment to support autonomous agents using hierarchical planners. The environment in turn generates multimodal evidence.
The authors provide two baselines against human performance: 1) Monte Carlo tree search with learned agent models and 2) GPT-4. Falling short against human performance, the authors observe different shortcomings of each baselines, highlighting the challenges of the benchmark.

---

> ### Author Rebuttal · Authors · 2024-08-16
>
> Thank you for your positive evaluation of our work and for your valuable feedback. We provide new LLM results in our attached PDF and clarification on the audio modality below. Please feel free to let us know if any further clarification is needed, and we would be happy to provide it.
>
> **Additional LLM Results**
>
> We completely agree that fine-tuning a weaker LLM based model could provide valuable insights, and we plan to address this in our future work.
>
> For now, we present new results (attached PDF, Figure 1) evaluating top state-of-the-art open-source LLMs (Llama-3.1-8B-Instruct and Qwen2-7B-Instruct) on our benchmark. We choose these models due to their large context length, as our prompt is over 11,000 tokens.
>
> Both LLMs struggle to perform the inference task. Llama-3.1’s performance is lower than but consistent with GPT-4’s. For scenarios where GPT-4 does converge, Llama-3.1 does not necessarily converge, but it shows an increase in inference accuracy as the trajectory progresses, indicating some signal. For scenarios where GPT-4 does not converge (“Who turned on the shower” and “Who turned on the laundry”), Llama-3.1’s inference accuracy does not improve with later evidence. We find that Llama-3.1 often reasons correctly about the state changes between timesteps, but it does not arrive at the correct conclusion. Meanwhile, Qwen2’s inference accuracy does not increase as the trajectory progresses and struggles to reason accurately about the state changes. We hope that our benchmark supports future work aimed at enhancing LLMs’ inference abilities.
>
>
> **Clarification on Audio Input**
>
> Audio input reveals partial information about the agent's low-level actions, and prior work shows that such auditory evidence can help resolve state uncertainty (Gerstenberg et al., 2021; Körding, 2007; Wu et al., 2024). Although we currently use audio input under a limited prediction setting, our motivation is to leverage it in ‘whodunit’ tasks under an inferential setting, which we plan to do in future work.
>
> For example, consider the scenario “Who turned on the laundry?” Suppose that visual evidence reveals that Agent A is in the same room as the laundry, just 5 steps away. Meanwhile, Agent B is in the bedroom, 20 steps away with the door closed. Based on just this, one might infer that Agent A was the likely culprit due to proximity. However, if the audio evidence reveals a long sequence of steps or a door closing, one might instead infer that agent B was responsible. This type of setup is explored in Wu et al., 2024, and we hope that our benchmark supports future work in this direction.
>
>
> **References**
>
> Gerstenberg, T., Siegel, M. H., & Tenenbaum, J. B. (2021). What happened? Reconstructing the past from vision and sound. PsyArXiv. https://psyarxiv.com/tfjdk
>
> Wu, S. A., Brockbank, E., Cha, H., Fränken, J.-P., Jin, E., Huang, Z., Liu, W., Zhang, R., Wu, J., & Gerstenberg, T. (2024). Whodunnit? Inferring what happened from multimodal evidence. In L. K. Samuelson, S. Frank, M. Toneva, A. Mackey, & E. Hazeltine (Eds.), Proceedings of the 46th Annual Conference of the Cognitive Science Society.
>
> Körding, K. P., Beierholm, U., Ma, W. J., Quartz, S., Tenenbaum, J. B., & Shams, L. (2007). Causal inference in multisensory perception. PLOS ONE, 2(9), e943.

---

> ### Author Response · Authors · 2024-08-26
> **Happy to answer any further questions**
>
> Dear Reviewer fKzP,
>
> Thank you for your detailed review our submission. We have put great effort into addressing your questions during the rebuttal period, specifically providing additional LLM results and clarification on the motivation of the audio input. We are excited to discuss with you and answer any further questions. Thank you again for taking the time to read our work, and we look forward to your feedback!
>
> Thank you,
>
> Authors

---

> ### Author Response · Authors · 2024-08-29
> **Looking forward to discussion**
>
> Dear Reviewer fKzP,
>
> Thank you again for your valuable feedback on our submission. We have posted our clarification and response, and we would be happy to hear your updated thoughts. We look forward to further discussions and are happy to answer any additional questions!
>
> Thank you,
>
> Authors

---

### Author Rebuttal · Authors · 2024-08-16

Thank you to all of our reviewers for their time and thoughtful comments and feedback on our work. Your insights have helped us with improving our baseline experiments, the clarity of our writing, and the documentation of our codebase. We have carefully addressed each reviewer's questions and comments individually and provided new results where recommended.

In our global response PDF, we provide the following new experimental results:
* [fKzP, AsPM] Additional experiments benchmarking open-source LLMs in Figure 1
* [shNX] Additional experiments on the effect of each modality on mental simulation baselines in Figures 2, 3

The following summarizes other key improvements that we have made based on the reviewers' suggestions:
* [fKzP, AsPM] Clarification on how the audio and language modalities contribute to inference
* [8BjY] Clarification on the long-horizon problem formulation and relevance of evidence
* [AsPM] Clarification on how our inference scenarios are motivated by the ‘Whodunit’ storyline
* [AsPM, shNX] Detailed documentation for the simulator and dataset in our [Github repository](https://github.com/marple-benchmark/marple)

We will incorporate these changes into our revised paper. We are eager to continue the discussion and are more than happy to provide any additional clarifications!

---

### Decision · Program_Chairs · 2024-09-26

**Decision:**

Accept (Poster)

**Comment:**

This paper presents a new multimodal benchmark for long horizon inference, posed as a series of “whodunit”-style reasoning problems. Utilising an existing simulator (Mini-BEHAVIOUR), the environment generates problems with visual, audio and text “evidence”, and an LLM must guess “who opened the door”/”who ate the sandwich” etc. The paper presents baselines including MCTS, LLM inference and benchmarks against human performance.

**Strengths:**

* Reviewers noted that the environment is comprehensive & customizable.
* Reviewers also noted the originality of the benchmark, and the motivation for the work.
* Reviewers favourably commented on the empirical analysis and demonstrated the limitations of current state of the art models in comparison to human-level performance.

**Weaknesses:**
* Baselines
    - Several reviewers suggested more baselines were required and perhaps with more rigorous justification. In response the authors provided more open source baselines and some more ablations of their Monte Carlo baselines.
* Questions about complexity and modality
    - Some questions were raised about the complexity and diversity of the simulated environment, and also the forms of the modalities raised (why audio? does text leak information?) and the authors engaged with the reviewers to answer these questions.

**AC Recommendation:**
Following the unanimous recommendation of the reviews, I recommend this for acceptance. Alongside the noted strengths listed the reviewers, I was pleased to see improvements of the paper over the review process (adding additional benchmarks) and in particular agree with the thorough review of shNX, who pointed out the originality of such a long-horizon reasoning task and the relative scarcity of comparable benchmarks.